# Three Forward, One Backward: Memory-Efficient Full-Rank Fine-Tuning of Large Models via Extra Forward Passes

**Jia Zhang**[1]*    **Yu Bai**[1]*    **Hualin Zhang**[2]*    **Tianshuo Chen**[1]*
**Zhaogeng Liu**[1]*    **Zhiqiang Xu**[2]    **Yi Chang**[1,3,4]    **Bin Gu**[1]†

[1]School of Artificial Intelligence, Jilin University, China
[2]Mohamed bin Zayed University of Artificial Intelligence, Abu Dhabi, UAE
[3]International Center of Future Science, Jilin University, China
[4]Engineering Research Center of Knowledge-Driven Human-Machine Intelligence, MOE, China
`{jiazhang25, baiy23, chents25, zgliu20}@mails.jlu.edu.cn`
`zhanghualin98@gmail.com`
`zhiqiang.xu@mbzuai.ac.ae`
`{yichang, gubin}@jlu.edu.cn`

## Abstract

Fine-tuning large language models (LLMs) has achieved significant success in downstream tasks. However, as the model size continues to grow, traditional fine-tuning methods have become increasingly impractical due to their high computational and memory costs. This has motivated researchers to explore parameter-efficient and memory-friendly fine-tuning strategies to enable scalable approaches, with Low-Rank Adaptation (LoRA) standing out as a representative work. However, the LoRA update is restricted to a low-rank subspace, which results in suboptimal performance compared to the full-parameter update. Recent research has also explored memory-efficient fine-tuning LLMs using just forward passes while suffer from high variance in gradient estimation and low convergence speed. To address the issues above, we propose a new alternating optimization framework called LMAO (**L**ow-rank and **M**emory-efficient Zeroth-Order **A**lternating **O**ptimization), which combines the advantages of LoRA and MeZO. This method alternately updates the low-rank components and zeroth-order directions during training. By performing three forward propagations and one backward propagation, each update is full-rank, thereby reducing feature loss and enabling efficient fine-tuning under strict memory constraints. We provide theoretical guarantees on the convergence and convergence rate of this method. Empirical results demonstrate that, in experiments on multiple models (e.g., OPT, RoBERTa-large), LMAO achieves performance comparable to first-order methods. This presents a practical and scalable solution for fine-tuning large-scale models. Our source code is available at https://github.com/workelaina/LMAO.

## 1 Introduction

Large language models (LLMs) have achieved remarkable performance across a wide range of domains (Solaiman et al., 2019; Brown et al., 2020; Achiam et al., 2023). However, as LLMs continue to scale, gradient computation can demand over $12\times$ the memory required for inference (Malladi et al., 2023), posing significant challenges to model training and fine-tuning — particularly exacerbating optimization difficulty in resource-constrained environments. This challenge has spurred the development of memory-efficient alternatives, such as parameter-efficient fine-tuning (PEFT), which aim to reduce resource requirements while preserving task-specific performance.

---

*Co-first authors with equal contributions.
†Corresponding author.

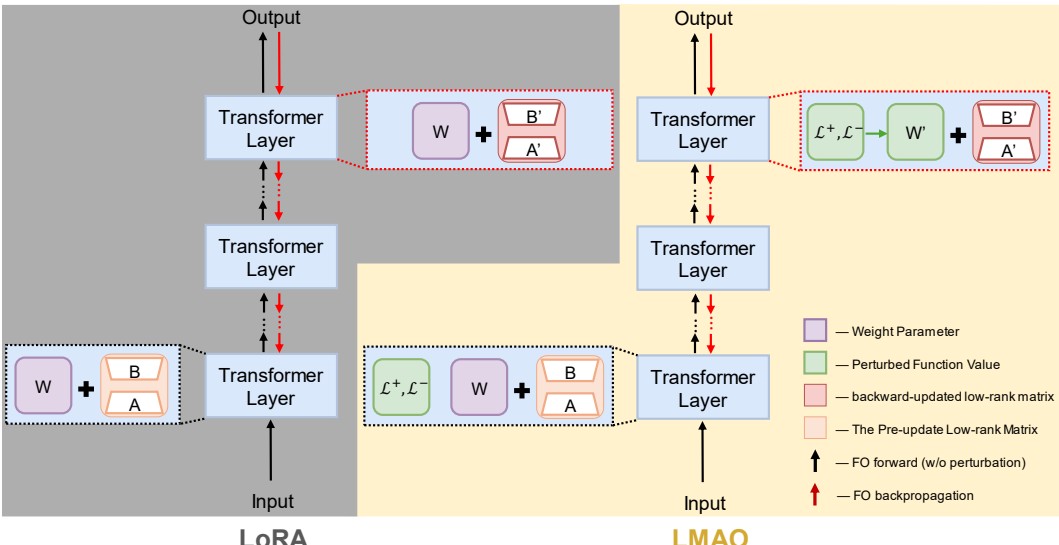

Figure 1: Comparison of standard LoRA and LMAO: LoRA performs a single forward-backward pass, while LMAO alternates it with two zeroth-order forward passes. Note: the forward and backward passes update only the low-rank matrices $B$ and $A$ in LoRA, while the base weight matrix $W$ is updated solely through the forward pass.

PEFT methods allow the adaptation of LLMs to downstream tasks by tuning only a subset of model parameters or by adding task-specific modules, such as adapters (Houlsby et al., 2019), prefix embeddings (Li & Liang, 2021), and prompt tuning (Lester et al., 2021), without altering the base architecture. Studies further demonstrate that fine-tuning pretrained models can be effectively conducted within reduced-dimensional parameter subspaces (Aghajanyan et al., 2020). Low-Rank Adaptation (LoRA), a prominent PEFT technique, leverages this insight by performing adaptations within optimized low-dimensional parameter spaces. This strategy significantly enhances the efficiency of adapting large-scale pretrained models, validating the theoretical foundations regarding low-rank representations in neural networks (Hu et al., 2022; Aghajanyan et al., 2020; Li et al., 2018). Despite the efficiency, LoRA still falls short of matching full-model fine-tuning performance due to its limited representation power of fine-tuned language models (Hu et al., 2022).

Meanwhile, zeroth-order optimization offers an alternative approach to model adaptation, particularly when gradient access is restricted. Zeroth-order optimizers approximate gradients through forward passes alone, eliminating the need for backpropagation and further reducing memory requirements. MeZO (Malladi et al., 2023), a Memory-efficient Zeroth-Order optimizer, established the feasibility of fine-tuning large models using forward passes only, enabling gradient-free adaptation for various downstream tasks. However, the inherent noise and high variance of zeroth-order gradient estimates pose limitations on convergence speed and task performance.

In this paper, we propose a memory-efficient full-rank fine-tuning method, the Low-rank and Memory-efficient Zeroth-order Alternating Optimization (LMAO) algorithm. LMAO combines LoRA and MeZO to optimize model performance by alternately updating low-rank matrix components and zeroth-order directions. Each iteration involves three forward passes and one backpropagation, ensuring full-rank updates. This approach reduces feature loss and enhances model expressiveness compared to traditional low-rank methods. LMAO also integrates the memory efficiency of both PEFT methods, ensuring tight control over memory usage. It introduces a new alternating optimization framework with theoretical convergence guarantees, making it a memory-efficient solution for fine-tuning large models. The algorithm is well-suited for memory-constrained environments, maintaining high performance while staying within memory limits, enabling efficient large-scale model optimization.

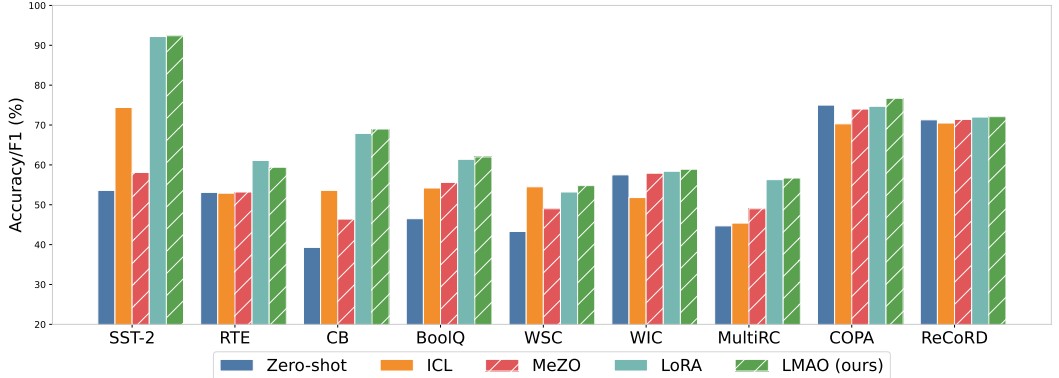

Figure 2: OPT-1.3B results under zero-shot, ICL, MeZO (Memory-efficient Zeroth-Order optimization), LoRA (Low-Rank Adaptation), and LMAO (ours). LMAO achieves superior performance on most tasks. See Table 2 for details.

## 2 RELATED WORK

**Parameter-Efficient Fine-Tuning.** Parameter-Efficient Fine-Tuning (PEFT) methods reduce the computational and memory costs of fine-tuning large pre-trained language models (LLMs). PEFT adapts LLMs to downstream tasks by adjusting a small subset of the model's parameters while keeping most weights fixed. This approach achieves task-specific improvements without the extensive hardware required for full-model fine-tuning. Prompt tuning (Lester et al., 2021) optimizes continuous prompt vectors appended to input embeddings, enabling task-specific conditioning. Prefix tuning (Li & Liang, 2021) introduces trainable prefix tokens at each transformer layer for lightweight conditioning. Adapters (Houlsby et al., 2019) add small neural modules between transformer layers, which are tuned independently. Low-Rank Adaptation (LoRA) (Hu et al., 2022) reduces parameter overhead by injecting low-rank matrices into attention and feedforward layers.

**Zeroth-Order Fine-Tuning of LLMs.** Zeroth-order (ZO) optimization methods enable gradient-free fine-tuning, ideal for scenarios where gradient access is infeasible or computationally expensive. MeZO (Malladi et al., 2023), a Memory-Efficient Zeroth-Order optimizer, shows that ZO fine-tuning can achieve memory efficiency similar to model inference with forward-pass-only updates. MeZO uses Simultaneous Perturbation Stochastic Approximation (SPSA) (Spall, 1992) for gradient estimation, allowing updates without backpropagation. However, MeZO faces challenges in convergence speed due to high variance in ZO gradient estimates. Recent improvements to MeZO focus on sparsity (Liu et al., 2024; Guo et al., 2024), variance reduction (Gautam et al., 2024), low-rank structures (Yu et al., 2024b; Chen et al., 2024), and Hessian information (Zhao et al., 2024; Yu et al., 2024a).

## 3 METHODOLOGY

### 3.1 PRELIMINARIES

**LoRA.** Low-Rank Adaptation of Large Language Models (Hu et al., 2022) is a classical fine-tuning method for large language models. By introducing trainable low-rank matrices, LoRA reduces the number of trainable parameters to less than 0.1% of the full model, decreases GPU memory usage by two-thirds, and introduces no additional inference latency. The study by Aghajanyan et al. (2020) demonstrates that pretrained models possess an extremely low intrinsic dimensionality, indicating that fine-tuning a small number of parameters in a low-dimensional subspace can achieve performance comparable to full-parameter fine-tuning. The core idea is that the parameter update $\Delta W$ with respect to the pretrained weight matrix $W_0 \in \mathbb{R}^{m \times n}$ can be represented via a low-rank decomposition, i.e.,

$$W_0 + \Delta W = W_0 + \frac{\alpha}{r} BA, \quad B \in \mathbb{R}^{m \times r}, \ A \in \mathbb{R}^{r \times n} \text{ and } r \ll \min(m, n).$$

**MeZO.** Although traditional ZO methods can estimate gradients using only two forward passes, they suffer from the burden of additional memory consumption induced by random perturbations,

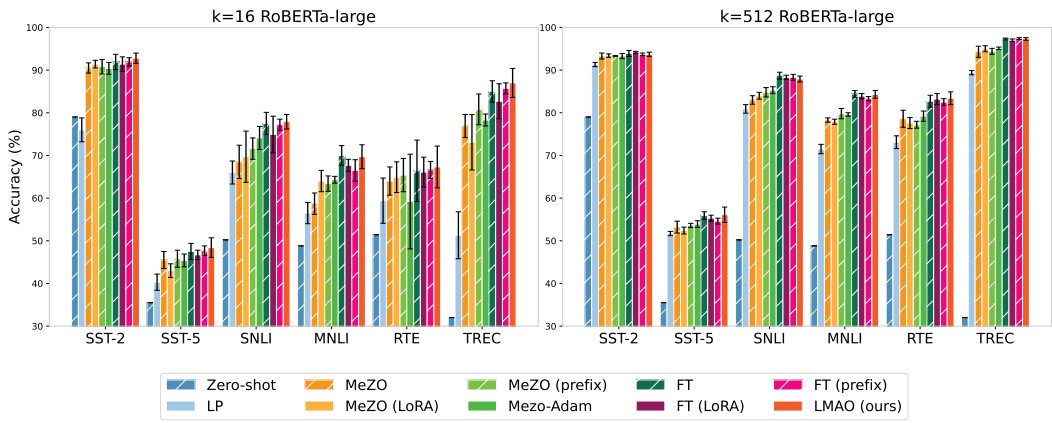

Figure 3: Experiments on RoBERTa-large. We report results for zero-shot, linear probing (LP), MeZO and its variants, our method LMAO, as well as full fine-tuning (FT), LoRA, and prefix-tuning. With $k = 16$, LMAO significantly outperforms LP and MeZO variants. At $k = 512$, LMAO continues to achieve strong performance. Detailed results are shown in Table 1.

which have the same shape as the model parameters. To address this limitation, Memory-efficient Zeroth-Order optimizer (MeZO) (Malladi et al., 2023) utilizes the random seed trick to generate random perturbations sequentially on-the-fly, enabling in-place optimization without modifying the model architecture and achieving memory usage equivalent to inference during fine-tuning. Moreover, MeZO exhibits broad compatibility, supporting both full-parameter fine-tuning and parameter-efficient techniques such as LoRA.

## 3.2 Low-rank and Memory-efficient Zeroth-Order Alternating Optimization

Although LoRA reduces trainable parameters by restricting updates to a low-dimensional subspace, this constraint limits the representational capacity of fine-tuned models, often yielding suboptimal performance compared to full-parameter updates (Hu et al., 2022). In contrast, MeZO avoids gradient computations but suffers from high-variance gradient estimates, leading to slower convergence (Yu et al., 2024a). To leverage the strengths of both, we propose the Low-rank and Memory-efficient zeroth-order Alternating Optimization (LMAO) framework, which strategically alternates between these complementary paradigms to enable efficient fine-tuning of LLMs under limited resources while maintaining strong performance.

Each LMAO training iteration consists of two phases: 1) Low-rank adaptation: Low-rank matrices $A$ and $B$ are updated using standard gradient-based optimizers (e.g., AdamW (Loshchilov & Hutter, 2017), SGD), involving a forward pass and a backward propagation to compute exact gradients for the low-rank modules. 2) Zeroth-order optimization: Following the low-rank update, a memory-efficient zeroth-order step adjusts the base model parameters $W$ using two perturbed forward passes to estimate gradients with respect to the global weights.

The full LMAO procedure is summarized in Algorithm 1. Each iteration performs three forward passes—one for the LoRA update and two perturbed passes for zeroth-order gradient estimation—and a single backward pass for LoRA gradients. This design ensures that global weight updates remain full-rank, substantially mitigating the feature loss inherent to purely low-rank or zeroth-order methods.

## 4 Theory

In this paper, we consider the following optimization problem:

$$\min_{W,A,B} \mathcal{L}(W; A, B), \tag{1}$$

where $W \in \mathbb{R}^{m \times n}$ represents the model parameters, $A \in \mathbb{R}^{r \times n}$, $B \in \mathbb{R}^{m \times r}$ are low-rank matrices, and $\mathcal{L}$ denotes the loss function.

---

**Algorithm 1** LMAO (Low-rank and Memory-efficient Zeroth-Order Alternating Optimization)

---

**Input:** Model Weight Parameters $W$, loss $\mathcal{L}$, step budget $T$, perturbation scale $\varepsilon$, batch size $\mathcal{B}$, learning rate scheduler $\{\eta_t\}$.

1: **for** $t = 1, \cdots, T$ **do**
2:      Sample batch $\mathcal{B} \subset \mathcal{D}$ and random seed $s$
3:      $[A, B] = \text{LoRA\_Update}(W, A, B)$                   ▷ Forward pass and backpropagation
4:      $W \leftarrow \text{PerturbParameters}(W, \varepsilon, s)$
5:      $\mathcal{L}^+ \leftarrow \mathcal{L}(W, A, B; \mathcal{B})$                       ▷ Forward pass w/ positive perturbation
6:      $W \leftarrow \text{PerturbParameters}(W, -2\varepsilon, s)$
7:      $\mathcal{L}^- \leftarrow \mathcal{L}(W, A, B; \mathcal{B})$                      ▷ Forward pass w/ negative perturbation
8:      $W \leftarrow \text{PerturbParameters}(W, \varepsilon, s)$
9:      $\text{projected\_grad} \leftarrow (\mathcal{L}^+ - \mathcal{L}^-)/(2\varepsilon)$
10:     Reset random number generator with seed $s$
11:     **for** $W_i \in W$ **do**
12:         $z \sim \mathcal{N}(0, 1)$
13:         $W_i \leftarrow W_i - \eta_t \cdot \text{projected\_grad} \cdot z$
14:     **end for**
15: **end for**
16: **function** $\text{LoRA\_Update}(W, A, B)$
17:     $[A, B] = [A, B] - \eta_t \nabla_{AB} \mathcal{L}(W, A, B; \mathcal{B})$
18:     **return** $[A, B]$
19: **end function**
20: **function** $\text{PerturbParameters}(W, \varepsilon, s)$
21:     Reset random number generator with seed $s$
22:     **for** $W_i \in W$ **do**
23:         $z \sim \mathcal{N}(0, 1)$
24:         $W_i \leftarrow W_i + \varepsilon z$
25:     **end for**
26:     **return** $W$
27: **end function**

---

We briefly introduce the theoretical foundations of the LoRA and MeZO components of our algorithm 1, followed by a theoretical analysis of their alternating optimization.

Firstly, we analyze the theoretical aspects based on the following assumptions.

**Assumption 4.1 (Lipschitz smoothness).** *Let $\mathcal{L}(W; A, B)$ be a differentiable function, there exist constants $L_W$ and $L_{BA}$ such that for any $W_1, W_2 \in \mathbb{R}^{m \times n}$, and $B_1 A_1, B_2 A_2 \in \mathbb{R}^{m \times n}$ the following condition holds:*

1. *$\|\nabla_W \mathcal{L}(W_1; A, B) - \nabla_W \mathcal{L}(W_2; A, B)\| \leqslant L_W \|W_1 - W_2\|,$*

2. *$\|\nabla_{BA} \mathcal{L}(W; A_1, B_1) - \nabla_{BA} \mathcal{L}(W; A_2, B_2)\| \leqslant L_{BA} \|[B_1, A_1] - [B_2, A_2]\|,$*

3. *$\|\nabla_W \mathcal{L}(W; A_1, B_1) - \nabla_W \mathcal{L}(W; A_2, B_2)\|_F^2 \leqslant L_{qua} \|B_1 A_1 - B_2 A_2\|_F^2.$*

**Assumption 4.2 (Expected smoothness (Khaled & Richtárik, 2020; Malinovsky et al., 2024)).** *The second moment of the stochastic gradient satisfies*

$$\mathbb{E}\left[\|\nabla \mathcal{L}(W; A, B)\|_F^2\right] \leqslant 2\alpha \left(\mathcal{L}(W; A, B) - \mathcal{L}^*\right) + \beta \|\nabla \mathcal{L}(W; A, B)\|_F^2$$

*for some $\alpha, \beta \geqslant 0$ and all $W \in \mathbb{R}^{m \times n}$, $B \in \mathbb{R}^{m \times r}$ and $A \in \mathbb{R}^{r \times n}$.*

**Part of LoRA.** We begin by discussing the LoRA component. The update procedure is as follows:

$$[B_{t+1}, A_{t+1}] = [B_t, A_t] - \eta_{BA} \nabla_{BA} \mathcal{L}(W_t; A_t, B_t), \tag{2}$$

where $\eta$ is the learning rate, and $\nabla_{BA} \mathcal{L}(W_t; A_t, B_t)$ represents the gradient as expressed below:

$$\nabla_{BA} \mathcal{L}(W_t; A_t, B_t) = [\nabla_B \mathcal{L}(W_t; A_t, B_t), \nabla_A \mathcal{L}(W_t; A_t, B_t)].$$

First, under Algorithm 1 and based on Assumption 4.1, we present the descent lemma for LMAO as follows.

**Lemma 4.3** (**Descent lemma**). *Let $\mathcal{L}$ be a function satisfying Assumption 4.1. Under the condition $\eta < 1/L_{\max}$, where $L_{\max} = \max(L_{BA}, L_W, L_{qua})$. Then we have*

$$\mathcal{L}(W_{t+1}; A_{t+1}, B_{t+1}) \leqslant \mathcal{L}(W_t; A_t, B_t)$$
$$- \frac{\eta}{2}\Big(\big\|\nabla_{BA}\mathcal{L}(W_t; A_t, B_t)\big\|_F^2 + \big\|\nabla_W\mathcal{L}(W_t; A_{t+1}, B_{t+1})\big\|_F^2\Big).$$

**Part of MeZO.** The MeZO method is an efficient memory optimization approach that combines zeroth-order methods. Given a labeled dataset $\mathcal{D}$ and a mini-batch $\mathcal{M}$ of size $|\mathcal{M}|$, we introduce the classic zeroth-order gradient as follows.

**Definition 4.4** (**Zeroth Gradient**). *Given a model with parameters $W \in \mathbb{R}^{m \times n}$ and loss function $\mathcal{L}$, SPSA approximates the gradient on a minibatch $\mathcal{M}$ as*

$$\hat{\nabla}_W\mathcal{L}(W; A, B; \mathcal{M}) = \frac{\mathcal{L}(W + \varepsilon z; A, B; \mathcal{M}) - \mathcal{L}(W - \varepsilon z; A, B; \mathcal{M})}{2\varepsilon}z \tag{3}$$
$$\approx zz^T\nabla_W\mathcal{L}(W; A, B; \mathcal{M}),$$

*where $z \in \mathbb{R}^{m \times n}$, $z \sim \mathcal{N}(0, I_{m \times n})$ and $\varepsilon \to 0^+$ is the perturbation scale.*

The update procedure of MeZO is as follows:

$$W_{t+1} = W_t - \eta_W\hat{\nabla}_W\mathcal{L}(W_t; A_{t+1}, B_{t+1}; \mathcal{M}). \tag{4}$$

**Definition 4.5.** *The SGD gradient estimate on a minibatch of size $\mathcal{M}$ has covariance $\sum_{\mathcal{M}}$ as*

$$|\mathcal{M}|\Big(\mathbb{E}\big[\nabla\mathcal{L}(W; A, B; \mathcal{M})\nabla^T\mathcal{L}(W; A, B; \mathcal{M})\big] - \nabla\mathcal{L}(W; A, B)\nabla^T\mathcal{L}(W; A, B)\Big).$$

$\hat{\nabla}_W\mathcal{L}(W; A, B; \mathcal{M})$ be a statistic, we need to examine the unbiasedness of its corresponding estimator, which leads to the following lemmas.

**Lemma 4.6** (**Unbiasedness**). *Based on Definition 4.4, for a mini-batch $\mathcal{M}$, we have*

$$\mathbb{E}_{z \sim \mathcal{N}(0,I)}\big[\hat{\nabla}_W\mathcal{L}(W; A, B; \mathcal{M})\big] = \hat{\nabla}_W\mathcal{L}(W; A, B).$$

**Lemma 4.7** (**Biased Estimate**). *Based on Definition 4.4, the gradient norm of MeZO is*

$$\mathbb{E}_{z \sim \mathcal{N}(0,I)}\big[\|\hat{\nabla}_W\mathcal{L}(W; A, B; \mathcal{M})\|_F^2\big] = \frac{mn + N - 1}{N}\mathbb{E}\big[\|\hat{\nabla}_W\mathcal{L}(W; A, B)\|_F^2\big],$$

*where $m, n$ is the size of parameters and $N$ is the number of $z$ sampled in SPSA (Definition 4.4).*

**Alternating Optimization.** In Algorithm 1, we optimize by alternately applying LoRA and MeZO methods. To analyze the convergence, we require the following lemma:

**Lemma 4.8.** *Let $\mathcal{L}(W; A, B)$ be the $L$-smoothness function, we have*

$$\|\nabla\mathcal{L}(W_t; A_t, B_t)\|_F^2 \leqslant 2\left(1 + \frac{L_{BA}}{L_{\max}}\right)^2\left(\|\nabla_{BA}\mathcal{L}(W_t; A_t, B_t)\|_F^2 + \|\nabla_W\mathcal{L}(W_t; A_{t+1}, B_{t+1})\|_F^2\right).$$

Under Assumptions 4.1 and Assumption 4.2, the convergence of Algorithm 1 is guaranteed by the following Theorem:

**Theorem 4.9.** *The iterates of Algorithm 1 with SGD satisfy*

$$\min_{0 \leqslant t \leqslant T-1}\mathbb{E}\left[\big\|\nabla\mathcal{L}(W_t; A_{t+1}, B_{t+1})\big\|_F^2\right] \leqslant \frac{6(1 + \frac{L_{BA}}{L_{\max}})^2(\mathcal{L}(W_0; A_0, B_0) - \mathcal{L}^*)}{\eta_{\max}T},$$

*where the stepsize $\eta_{\max} := \max(\eta_{BA}, \eta_W)$ satisfy*

$$\eta_{\max} \leqslant \min\bigg(\frac{N}{1000\beta_2 L_W(mn + N - 1)}, \frac{1000N}{N\beta_1 L_{BA} + 2\alpha_2 L_W(mn + N - 1)},$$
$$\frac{4}{L_{BA}}, \sqrt{\frac{N}{T(N\alpha_1 L_{BA} + \alpha_2 L_W(mn + N - 1))}}\bigg)$$

*and is set according to the experimental configuration as specified in expression $\frac{\eta_{\min}}{\eta_{\max}} \geqslant \frac{1}{1000}$.*

Building on the finite series construction technique introduced in (Malinovsky et al., 2024) for the theoretical analysis of LoRA, we incorporate SGD-based zeroth-order optimization and tangential scaling inequalities in an alternating scheme (Lu et al., 2019). This leads to an adaptive convergence analysis, with detailed proofs provided in the appendix.

## 5 EXPERIMENTS

In this section, we present comprehensive experiments on multiple fine-tuning tasks across models with different parameter scales, aiming to rigorously evaluate the performance of LMAO. The experimental setup is detailed as follows:

**Models.** The experiments are evaluated across multiple language models, including the medium-sized RoBERTa-large(Liu et al., 2019) and the large autoregressive Open Pre-trained Transformers (OPT) series(Zhang et al., 2022), comprising OPT-1.3B, OPT-2.7B, and OPT-6.7B.

**Datasets.** We conduct experiments across multiple fine-tuning tasks in GLUE (Wang et al., 2018), SuperGLUE (Wang et al., 2019) benchmark and other datasets. For RoBERTa-large, we conduct experiments on SST-2 (Socher et al., 2013), SST-5 (Socher et al., 2013), SNLI (Bowman et al., 2015), MNLI (Williams et al., 2017), RTE (Dagan et al., 2005; Haim et al., 2006; Giampiccolo et al., 2007; Bentivogli et al., 2009), and TREC (Voorhees & Tice, 2000). For OPT-1.3B, we conduct experiments on SST-2 (Socher et al., 2013), CB (De Marneffe et al., 2019), BoolQ (Clark et al., 2019), WSC (Levesque et al., 2012), WIC (Pilehvar & Camacho-Collados, 2018), MultiRC (Khashabi et al., 2018), COPA (Roemmele et al., 2011), and ReCoRD (Zhang et al., 2018).

**Baselines.** The baselines compared in our experiments contain both gradient-free-based and gradient-based methods, including: 1) Zero-shot, 2) In-context learning, 3) Memory-efficient zeroth-order optimizer (MeZO), 4) Low-rank adaptation (LoRA), and 5) Fine-tuning (Full-parameter). The available code for the baselines is from https://github.com/princeton-nlp/MeZO.

### 5.1 MEDIUM-SIZED MASKED LANGUAGE MODELS

We first conduct experiments using the masked language model RoBERTa-large. We compare the performance of the LMAO with other methods on sentiment analysis, natural language inference, and topic classification tasks. We adopt the classic approaches from previous studies, including few-shot and many-shot settings. For $k = 16$ and $k = 512$, we sample $k$ cases per class. A uniform fine-tuning of $1K$ steps is applied. The results are summarized in Table 1 and Figure 3.

**LMAO significantly outperforms Zero-shot, LP, MeZO, and its variants.** Across all six tasks, LMAO consistently optimizes the pre-trained model and outperforms Zero-shot, LP, MeZO, and its various extensions. On several tasks, it even surpasses full fine-tuning (FT), with the remaining performance gaps within 0.5%.

**LMAO approaches the performance of full fine-tuning (FT) as the number of samples increases, with at most a 2% gap.** For $k = 16$, LMAO consistently outperforms LoRA across all six tasks, and in most cases even surpasses FT. When increasing $k$, the performance gap between LMAO and FT narrows further, with some tasks showing less than a 1% difference.

### 5.2 LARGE-SCALE AUTOREGRESSIVE LANGUAGE MODELS

After achieving promising results with RoBERTa-large, we extend the LMAO approach to the OPT series models, including three scales with 1.3B, 2.7B, and 6.7B parameters. To systematically evaluate the performance of LMAO, we select several tasks from the SuperGLUE (Wang et al., 2019) benchmark, encompassing text classification, multiple-choice, and generation tasks. In the experiments, we compare LMAO with several state-of-the-art methods, including LoRA, MeZO, ICL (In-context Learning), and Zero-shot learning. To assess the performance differences under identical training conditions, all experiments are conducted with consistent settings, with each model fine-tuned for $1K$ iterations. This setup allows for a more accurate reflection of the strengths and weaknesses of each method. The results for OPT-1.3B are summarized in Table 2 and Figure 2, while additional results for larger models are provided in Appendix A.

Table 1: Experiments on RoBERTa-large (350M parameters). LP: Linear probing; ZO, ZO (LoRA), and ZO (prefix): memory-efficient ZO-SGD with full-parameter tuning, LoRA, and prefix-tuning respectively; FT: fine-tuning with Adam. All reported numbers are averaged accuracy (standard deviation). Bold numbers indicate the best performance on each task, while underlined numbers denote the second-best.

| Task | SST-2 | SST-5 | SNLI | MNLI | RTE | TREC |
|------|-------|-------|------|------|-----|------|
| Type | — sentiment — | | — natural language inference — | | | – topic – |
| Zero-shot | 79.0 | 35.5 | 50.2 | 48.8 | 51.4 | 32.0 |
| **Gradient-free methods:** $k = 16$ | | | | | | |
| LP | 76.0 (2.8) | 40.3 (1.9) | 66.0 (2.7) | 56.5 (2.5) | 59.4 (5.3) | 51.3 (5.5) |
| MeZO | 90.5 (1.2) | 45.5 (2.0) | 68.5 (3.9) | 58.7 (2.5) | 64.0 (3.3) | 76.9 (2.7) |
| MeZO (LoRA) | 91.4 (0.9) | 43.0 (1.6) | 69.7 (6.0) | 64.0 (2.5) | 64.9 (3.6) | 73.1 (6.5) |
| MeZO (prefix) | 90.8 (1.7) | 45.8 (2.0) | 71.6 (2.5) | 63.4 (1.8) | 65.4 (3.9) | 80.8 (3.6) |
| MeZO-Adam | 90.4 (1.4) | 45.4 (1.5) | 74.1 (2.7) | 64.3 (0.8)† | 59.2 (11.1)† | 78.3 (1.4) |
| **Gradient-based methods:** $k = 16$ | | | | | | |
| FT | 91.9 (1.8) | 47.5 (1.9) | 77.5 (2.6) | **70.0 (2.3)** | 66.4 (7.2) | 85.0 (2.5) |
| FT (LoRA) | 91.4 (1.7) | 46.7 (1.1) | 74.9 (4.3) | 67.7 (1.4) | 66.1 (3.5) | 82.7 (4.1) |
| FT (prefix) | 91.9 (1.0) | 47.7 (1.1) | 77.2 (1.3) | 66.5 (2.5) | 66.6 (2.0) | 85.7 (1.3) |
| **ours:** $k = 16$ | | | | | | |
| LMAO (ours) | **92.8 (1.2)** | **48.4 (2.3)** | **77.9 (1.7)** | 69.7 (2.8) | **67.3 (4.9)** | **87.0 (3.4)** |
| **Gradient-free methods:** $k = 512$ | | | | | | |
| LP | 91.3 (0.5) | 51.7 (0.5) | 80.9 (1.0) | 71.5 (1.1) | 73.1 (1.5) | 89.4 (0.5) |
| MeZO | 93.3 (0.7) | 53.2 (1.4) | 83.0 (1.0) | 78.3 (0.5) | 78.6 (2.0) | 94.3 (1.3) |
| MeZO (LoRA) | 93.4 (0.4) | 52.4 (0.8) | 84.0 (0.8) | 77.9 (0.6) | 77.6 (1.3) | 95.0 (0.7) |
| MeZO (prefix) | 93.3 (0.1) | 53.6 (0.5) | 84.8 (1.1) | 79.8 (1.2) | 77.2 (0.8) | 94.4 (0.7) |
| MeZO-Adam | 93.3 (0.6) | 53.9 (0.8) | 85.3 (0.8) | 79.6 (0.4) | 79.2 (1.2) | 95.1 (0.3) |
| **Gradient-based methods:** $k = 512$ | | | | | | |
| FT | 93.9 (0.7) | 55.9 (0.9) | **88.7 (0.8)** | **84.4 (0.8)** | 82.7 (1.4) | 97.3 (0.2) |
| FT (LoRA) | **94.2 (0.2)** | 55.3 (0.7) | 88.3 (0.5) | 83.9 (0.6) | 83.2 (1.3) | 97.0 (0.3) |
| FT (prefix) | 93.7 (0.3) | 54.6 (0.7) | 88.3 (0.7) | 83.3 (0.5) | 82.5 (0.8) | **97.4 (0.2)** |
| **ours:** $k = 512$ | | | | | | |
| LMAO (ours) | 93.7 (0.5) | **56.1 (1.8)** | 87.9 (0.7) | 84.3 (0.9) | **83.4 (1.5)** | 97.3 (0.3) |

Table 2: Experiments on OPT-1.3B (with 1000 examples). ICL: in-context learning; LP: linear probing; LoRA: Low-Rank Adaptation; LMAO: Low-rank and Memory-efficient Zeroth-Order Alternating Optimization. LMAO achieves the best performance on 8 out of 9 tasks and ranks second only to LoRA on RTE, with a marginal difference. Bold numbers indicate the best performance on each task, while underlined numbers denote the second-best.

| Method | Tasks | | | | | | | | |
|--------|-------|-----|-----|-------|-----|-----|---------|------|--------|
| | SST-2 | RTE | CB | BoolQ | WSC | WIC | MultiRC | COPA | ReCoRD |
| | ——————————— classification ——————————— | | | | | | | – multiple choice – | |
| Zero-shot | 53.6 | 53.1 | 39.3 | 46.5 | 43.3 | 57.5 | 44.7 | 75.0 | 71.3 |
| ICL | 74.4 | 52.9 | 53.6 | 54.2 | 54.5 | 51.8 | 45.4 | 70.3 | 70.5 |
| MeZO | 58.1 | 53.2 | 46.4 | 55.6 | 49.0 | 57.9 | 49.0 | 74.0 | 71.4 |
| LoRA | 92.2 | **61.1** | 67.9 | 61.4 | 53.2 | 58.4 | 56.3 | 74.7 | 72.0 |
| LMAO (ours) | **92.4** | 59.4 | **69.0** | **62.0** | **54.8** | **58.9** | **56.7** | **76.7** | **72.1** |

**LMAO combines efficiency with strong generalization capabilities.** On the OPT-1.3B model, it achieves superior performance across a range of SuperGLUE tasks, consistently outperforming mainstream fine-tuning and zero-shot approaches, positioning it as a competitive alternative framework.

**LMAO (ours) achieves the best performance on the majority of tasks.** LMAO ranks first on 7 out of 9 benchmarks, including substantial gains on SST-2, CB, BoolQ, COPA, and ReCoRD. On the remaining two tasks, its performance is competitive, highlighting the robustness and consistency of the approach.

**LMAO also serves as a strong PEFT approach.** While LoRA is a widely adopted PEFT baseline with competitive performance, LMAO consistently outperforms it on 8 out of 9 tasks, establishing itself as a high-performing and promising PEFT paradigm.

**LMAO effectively combines the advantages of zeroth-order optimization and low-rank adaptation.** Compared to zero-shot, ICL, and the zeroth-order method MeZO, LMAO demonstrates consistently superior performance. While MeZO generally outperforms Zero-shot and ICL, it still lags behind LMAO—for instance, MeZO achieves comparable results on WiC (57.9% vs. 58.9%) and ReCoRD (71.4% vs. 72.1%), but exhibits notable gaps on other tasks.

## 5.3 ABLATION STUDY

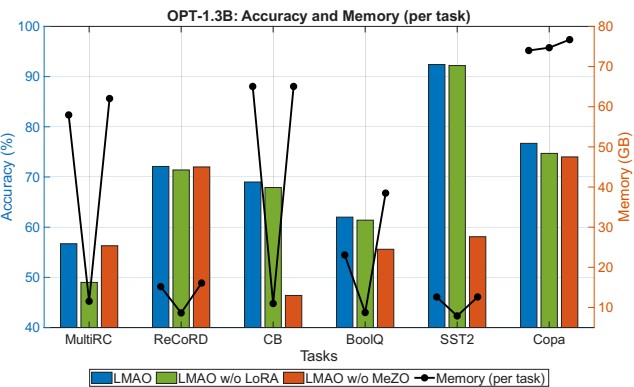

Figure 4: Ablation study results.

We conduct comprehensive ablation studies to systematically assess the effectiveness of our method. All experiments are performed on multiple benchmark tasks using the OPT-1.3B model. The ablation analysis examines several key components, including:

- **LMAO w/o LoRA:** Freezing the low-rank LoRA blocks optimized by LMAO, and updating only the zero-order optimizer components.
- **LMAO w/o MeZO:** Freezing the zero-order optimizer components optimized by LMAO, and updating only the low-rank LoRA blocks.

Figure 4 presents the ablation results across several representative tasks. As shown, the full LMAO method consistently achieves substantially higher accuracy than its individual components, namely the low-rank LoRA module and the zeroth-order MeZO module. This indicates that the integration of these two update mechanisms enables LMAO to more effectively capture task-relevant information and enhances the model's adaptation capability. The performance gains are observed across all evaluated datasets, further validating the effectiveness and robustness of the alternating optimization strategy adopted in LMAO.

Notably, despite combining two distinct update paradigms within a unified optimization framework, LMAO does not incur higher peak memory consumption compared to using LoRA or MeZO alone. This indicates that the integration of low-rank adaptation and zeroth-order updates is designed without introducing redundant intermediate states or additional backward-pass overhead. Consequently, LMAO achieves improved performance without adding memory overhead, maintaining a memory footprint comparable to its component methods. Overall, these findings highlight that LMAO delivers superior task performance while preserving computational efficiency and resource usage, underscoring its practical value in memory-constrained training scenarios.

## 5.4 PARAMETER SENSITIVITY ANALYSIS

In our method, $r$ and $\alpha$ are two key hyperparameters: $r$ determines the rank of the trainable low-rank matrices, while $\alpha$ scales the contribution of the adapter's output during fine-tuning. To comprehensively analyze the sensitivity of parameters w.r.t. $r$ and $\alpha$, we evaluate the RoBERTa-large model on the SST-2 dataset with varying values of $r$ and $\alpha$. The results are provided in Figure 5. We can see that our method is not sensitive to the settings of $r$ and $\alpha$. Therefore, in practical applications, we can prioritize smaller values of $r$ to ensure fine-tuning of the model with as few parameters as possible.

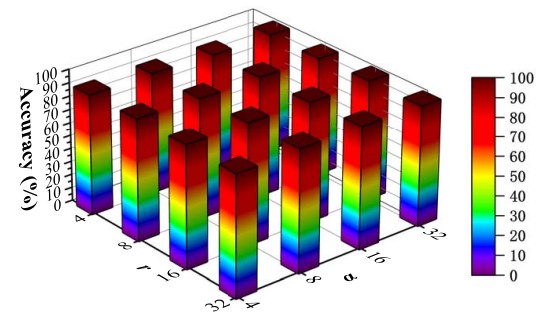

Figure 5: Parameter sensitivity w.r.t. hyperparameters $r$ and $\alpha$ in LMAO.

## 5.5 MEMORY UTILIZATION ANALYSIS

To examine the memory footprint of LMAO, we measure the peak GPU memory consumption on a subset of datasets (CB, SST-2, BoolQ, and COPA) and compare it against two representative baselines: the low-rank adaptation method LoRA and the zeroth-order optimization method MeZO. To ensure a fair comparison, all methods are evaluated under identical training configurations and hardware settings. The statistics are summarized in Table 3.

Table 3: Peak GPU Memory (GB) Usage on OPT-1.3B (1000 Examples). LoRA: Low-Rank Adaptation; MeZO: Memory-Efficient Zeroth-Order Optimizer; LMAO: Low-rank and Memory-efficient Zeroth-Order Alternating Optimization.

| Method | SST-2 | CB | BoolQ | COPA |
|---|---|---|---|---|
| MeZO | 7.90 | 10.99 | 8.75 | 7.70 |
| LoRA | 12.62 | 65.05 | 38.48 | 8.31 |
| LMAO (ours) | 12.62 | 65.05 | 23.08 | 8.31 |

As shown in Table 3, the peak memory consumption of LMAO never exceeds that of LoRA or MeZO, despite integrating both low-rank updates and zeroth-order optimization mechanisms. This indicates that LMAO does not introduce additional memory burden during training and maintains a memory footprint comparable to its component methods. Such efficiency is particularly important in resource-constrained settings, demonstrating that LMAO can be deployed without increasing hardware requirements.

More importantly, LMAO exhibits substantial performance gains across a variety of tasks, consistently outperforming both LoRA and MeZO on classification and multiple-choice benchmarks. These results suggest that LMAO effectively leverages the complementary strengths of the two update strategies, enhancing the model's learning capability and adaptability while preserving computational efficiency. Overall, the findings strongly support the design philosophy of LMAO, showing that it delivers notable performance improvements while remaining memory-efficient.

## 6 CONCLUSION

We introduce LMAO, a memory-efficient fine-tuning method for large language models that leverages full-rank updates. Unlike traditional low-rank adaptation (e.g., LoRA) and zeroth-order optimization (e.g., MeZO) techniques, which face inherent limitations, LMAO combines the strengths of both approaches by alternating low-rank matrix updates and zeroth-order forward-pass updates. This hybrid design significantly enhances model performance while effectively maintaining strict memory consumption, making it particularly suitable for large language models under resource constraints. Our theoretical analysis establishes rigorous convergence guarantees, ensuring the robustness and reliability of the method. Empirical experiments conducted on widely-used models such as RoBERTa-large and OPT demonstrate that LMAO delivers competitive performance compared to existing baseline methods. A remaining challenge lies in its training inefficiency on large datasets or long sequences. We propose that this issue could be addressed through optimization techniques such as component training, which remains an important avenue for future research and development.

## ACKNOWLEDGMENTS

Dr. Zhaogeng Liu was supported by the Young Scientists Fund (C Class) of the National Natural Science Foundation of China under Grant No. 62506142. Dr. Yi Chang was supported by the National Key R&D Program of China under Grant No. 2023YFF0905400, and the National Natural Science Foundation of China through grant No. U2341229.

## ETHICS STATEMENT

All participants in this work, as well as the paper submission, adhere to the ICLR Code of Ethics (https://iclr.cc/public/CodeOfEthics).

## REPRODUCIBILITY STATEMENT

We affirm that the results of this work are fully reproducible. Appendix C provides the theoretical proofs. Appendix A details the experimental implementations. The source code is available at https://github.com/workelaina/LMAO.

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

APPENDIX

# A EXPERIMENT SETUP

## A.1 DATASETS

For the RoBERTa-large experiments, we consider a set of standard classification benchmarks, including SST-2 (Socher et al., 2013), SST-5 (Socher et al., 2013), MNLI (Williams et al., 2017), SNLI (Bowman et al., 2015), and RTE (Dagan et al., 2005; Haim et al., 2006; Giampiccolo et al., 2007; Bentivogli et al., 2009). To enable efficient evaluation, we follow the setting in (Malladi et al., 2023) and limit the test set to 1000 examples. For training and validation, we adopt two configurations with $k = 16$ and $k = 512$, where $k$ denotes the number of examples per class for each split.

For the OPT experiments, we evaluate on the SuperGLUE benchmark suite (Wang et al., 2019), which comprises BoolQ (Clark et al., 2019), CB (De Marneffe et al., 2019), COPA (Roemmele et al., 2011), MultiRC (Khashabi et al., 2018), ReCoRD (Zhang et al., 2018), RTE (Dagan et al., 2005; Haim et al., 2006; Giampiccolo et al., 2007; Bentivogli et al., 2009), WIC (Pilehvar & Camacho-Collados, 2018), and WSC (Levesque et al., 2012). In addition, we include SST-2 (Socher et al., 2013). For all tasks, we randomly sample 1000 examples for training, 500 for validation, and 1000 for testing.

## A.2 PROMPTS

LMAO adopts the same zeroth-order technique as (Malladi et al., 2023), and therefore we follow their setup for the downstream tasks and prompt templates used to fine-tune RoBERTa-large. The specific templates are provided in Table 4.

Table 4: The prompts used in our RoBERTa-large experiments (see Table 1 and Figure 2) follow the setup in (Malladi et al., 2023), including a template and a set of label words that fill the [MASK] token. The symbols $< S_1 >$ and $< S_2 >$ denote the first and second input sentences, respectively.

| Datasets | C | Type | Prompt | Label words |
|---|---|---|---|---|
| SST-2 | 2 | sentiments cls. | $< S_1 >$ It was [MASK]. | {great, terrible} |
| SST-5 | 5 | sentiments cls. | $< S_1 >$ It was [MASK]. | {great, good, okay, bad, terrible} |
| TREC | 6 | topic cls. | [MASK] : $< S_1 >$ . | {Description, Expression, Entity, Human, Location, Number} |
| MNLI | 3 | NLI | $< S_1 >$? [MASK], $< S_2 >$ | {Yes, Maybe, No} |
| SNLI | 3 | NLI | $< S_1 >$? [MASK], $< S_2 >$ | {Yes, Maybe, No} |
| RTE | 2 | NLI | $< S_1 >$? [MASK], $< S_2 >$ | {Yes, No} |

Table 5 presents the prompt templates used in our OPT experiments. Specifically, the OPT setup includes two types of tasks: classification and multiple-choice. All prompts follow the setup in (Socher et al., 2013).

## A.3 HYPERPARAMETERS

For experiments with the RoBERTa-large model, we follow the prompt-based fine-tuning paradigm for masked language models as described in (Socher et al., 2013). In OPT experiments, we adopt a similar training strategy to (Socher et al., 2013) for classification tasks, where we extract logits corresponding to the label words and apply cross-entropy loss. For multiple-choice and generative tasks (e.g., QA), we retain only the correct candidate and use teacher forcing on the correct examples. Loss is computed solely on the tokens in the candidate portion, excluding the prompt.

The specific hyperparameter settings for RoBERTa-large and OPT are provided in Table 6 and Table 7, respectively.

Table 5: We use prompt templates in our OPT model experiments, covering classification (cls.) and multiple-choice (mch.) tasks. Templates follow (Malladi et al., 2023). For multiple-choice, we score candidates by average log-likelihood and select the highest. For QA, we apply greedy decoding.

| Dataset | Type | Prompt |
|---|---|---|
| SST-2 | cls. | <text >It was terrible/great |
| RTE | cls. | <preamise >
Does this mean that '<hypothesis>'is true? Yes or No?
Yes/No/Maybe |
| CB | cls. | Suppose <premise>Can we infer that '<hypothesis>'? Yes, No, or Maybe?
Yes/No/Maybe |
| BoolQ | cls. | <passage><question>
Yes/No |
| WSC | cls. | <text>
Does the word '<word>'have the same meaning in
these two sentences? Yes, No?
Yes/No |
| WIC | cls. | Does the word '<word>'have the same meaning in
these two sentences? Yes, No?
<sent1>
<sent2>
Yes/No |
| MultiRC | cls. | <paragraph>
Question: <question>
I found this answer '<answer>'. Is that correct? Yes or No?
Yes/No |
| COPA | mch. | <premise>so/because <candidate> |
| ReCoAD | mch. | <passage>
<query>.replace('@placeholder', <candidate>) |

Table 6: Main hyperparameter settings for all tasks (SST-2, RTE, CB, BoolQ, WSC, WIC, MultiRC, COPA, ReCoRD) under the RoBERTa-large model.

| Method | Batch size | Learning rate $\eta_{BA}$ | Learning rate $\eta_W$ | Scaling factor $\alpha$ | Rank $r$ | Perturbation scale $\varepsilon$ |
|---|---|---|---|---|---|---|
| LMAO | 64 | 3e-4 | 1e-7 | 24 | 8 | 1e-3 |
| LoRA | {4, 8, 16} | {1e-4, 3e-4, 5e-4} | / | 16 | 8 | / |
| FT (LoRA) | {4, 8, 16} | {1e-4, 3e-4, 5e-4} | / | 16 | 8 | / |
| MeZO | 64 | / | {1e-7, 1e-6, 1e-5} | / | / | 1e-3 |
| Zero-shot | 64 | / | / | / | / | / |

# B  MORE EXPERIMENT RESULTS

## B.1  PERFORMANCE ON LARGER MODELS

We scale up to the OPT-2.7B and OPT-6.7B model to evaluate LMAO and compare it with several state-of-the-art methods, including LoRA, MeZO, ICL (in-context learning), and zero-shot learning. All methods are trained with 1000 iterations under consistent settings. The detailed results are shown in Table 8 and Table 9.

Results show that LMAO performs strongly on OPT-2.7B, achieving superior results on six tasks (CB, BoolQ, WIC, MultiRC, COPA, ReCoRD). We further scale the model to OPT-6.7B and evaluate algorithm performance against MeZO and LoRA on a subset of tasks. The results show that LMAO continues to perform excellently on OPT-6.7B, achieving the best performance on four out of six tasks (CB, BoolQ, WIC, MultiRC, COPA, ReCoRD). Combined with the results on OPT-2.7B, this demonstrates LMAO's robustness and cross-task generalization, validating its effectiveness and adaptability on larger models. Thus, LMAO proves to be an efficient and memory-friendly fine-tuning approach for large language models.

Table 7: Hyperparameter settings for all datasets and corresponding methods under the OPT-1.3B model.

| Method | Hyperparameters | Tasks | | | | | | | | |
|---|---|---|---|---|---|---|---|---|---|---|
| | | SST-2 | RTE | CB | BoolQ | WSC | WIC | MultiRC | COPA | ReCoRD |
| LMAO | Batch size | 8 | 8 | 8 | 2 | 8 | 8 | 2 | 8 | 2 |
| | Learning rate $\eta_{BA}$ | 1e-5 | 1e-5 | 1e-5 | 1e-5 | 1e-5 | 1e-5 | 1e-5 | 1e-5 | 1e-5 |
| | Learning rate $\eta_W$ | 1e-6 | 1e-6 | 1e-6 | 1e-6 | 1e-6 | 1e-6 | 1e-6 | 1e-6 | 1e-6 |
| | Scaling factor $\alpha$ | 16 | 16 | 16 | 16 | 16 | 16 | 16 | 16 | 16 |
| | rank $r$ | 8 | 8 | 8 | 8 | 8 | 8 | 8 | 8 | 8 |
| | Perturbation scale $\varepsilon$ | 1e-3 | 1e-3 | 1e-3 | 1e-3 | 1e-3 | 1e-3 | 1e-3 | 1e-3 | 1e-3 |
| LoRA | Batch size | 8 | 8 | 8 | 2 | 8 | 8 | 2 | 8 | 2 |
| | Learning rate $\eta$ | 1e-5 | 1e-5 | 1e-5 | 1e-5 | 1e-5 | 1e-5 | 1e-5 | 1e-5 | 1e-5 |
| | Scaling factor $\alpha$ | 16 | 16 | 16 | 16 | 16 | 16 | 16 | 16 | 16 |
| | rank $r$ | 8 | 8 | 8 | 8 | 8 | 8 | 8 | 8 | 8 |
| MeZO | Batch size | 8 | 8 | 8 | 2 | 8 | 8 | 2 | 8 | 2 |
| | Learning rate $\eta$ | 1e-7 | 1e-7 | 1e-7 | 1e-7 | 1e-7 | 1e-7 | 1e-7 | 1e-7 | 1e-7 |
| | Perturbation scale $\varepsilon$ | 1e-3 | 1e-3 | 1e-3 | 1e-3 | 1e-3 | 1e-3 | 1e-3 | 1e-3 | 1e-3 |
| ICL | Batch size | 32 | 32 | 32 | 32 | 32 | 32 | 32 | 32 | 32 |
| Zero-shot | Batch size | 64 | 64 | 64 | 64 | 64 | 64 | 64 | 64 | 64 |
| LOZO | Batch size | 16 | / | 16 | 16 | / | / | / | 16 | / |
| | rank r | 8 | / | 8 | 8 | / | / | / | 8 | / |
| | Learning rate | 1e-5 | / | 1e-5 | 1e-5 | / | / | / | 1e-5 | / |
| SubZero | Batch size | 16 | / | 16 | 16 | / | / | / | 16 | / |
| | Learning rate | 1e-5 | / | 1e-5 | 1e-5 | / | / | / | 1e-5 | / |
| DoRA | Batch size | 16 | / | 16 | 16 | / | / | / | 16 | / |
| | rank r | 8 | / | 8 | 8 | / | / | / | 8 | / |
| | Learning rate | 1e-5 | / | 1e-5 | 1e-5 | / | / | / | 1e-5 | / |
| AdaLoRA | Batch size | 16 | / | 16 | 16 | / | / | / | 16 | / |
| | rank r | 8 | / | 8 | 8 | / | / | / | 8 | / |
| | Learning rate | 1e-5 | / | 1e-5 | 1e-5 | / | / | / | 1e-5 | / |

Table 8: Results on OPT-2.7B (with 1,000 examples). ICL: in-context learning; MeZO: Memory-efficient Zeroth-Order optimizer; LoRA: Low-Rank Adaptation; LMAO: Low-rank and Memory-efficient zeroth-order Alternating Optimization. LMAO achieves the best performance across all six tasks. Bold values denote the best result per task.

| Method | Tasks | | | | | |
|---|---|---|---|---|---|---|
| | CB | BoolQ | WIC | MultiRC | COPA | ReCoRD |
| | classification | | | | – multiple choice – | |
| Zero-shot | 50.0 | 50.9 | 52.5 | 43.2 | 71.0 | 75.0 |
| ICL | 55.4 | 56.9 | 51.6 | 52.4 | 77.0 | 74.0 |
| MeZO | 56.5 | 59.5 | 52.4 | 48.2 | 73.0 | 74.9 |
| LoRA | 66.1 | 68.3 | 55.5 | 56.4 | 83.3 | 75.2 |
| LMAO (ours) | **68.5** | **68.5** | **57.3** | **57.5** | **85.3** | **75.3** |

This demonstrates LMAO's robustness and cross-task generalization, validating its effectiveness and adaptability on larger models. LMAO thus serves as an efficient and memory-friendly fine-tuning approach for large language models.

## B.2 COMPARISON WITH MORE METHODS

To enable a more rigorous comparison with recent parameter-efficient fine-tuning (PEFT) approaches, we augment our evaluation with additional zeroth-order and LoRA variants. The experimental results are summarized in Table 10. Based on Table 10, LMAO attains the best overall performance across

Table 9: Results on OPT-6.7B (with 1,000 examples). MeZO: Memory-efficient Zeroth-Order optimizer; LoRA: Low-Rank Adaptation; LMAO: Low-rank and Memory-efficient zeroth-order Alternating Optimization. LMAO leads on seven of nine tasks. Bold values denote the best result per task.

| Method | Tasks | | | | | |
|---|---|---|---|---|---|---|
| | **CB** | **BoolQ** | **WIC** | **MultiRC** | **COPA** | **ReCoRD** |
| | ————— classification ————— | | | | – multiple choice – | |
| MeZO | 50.0 | 44.2 | 52.7 | 49.9 | 80.3 | 78.1 |
| LoRA | 64.3 | 49.4 | **52.9** | 57.0 | 81.0 | **79.7** |
| LMAO (ours) | **66.1** | **65.1** | 50.0 | **57.2** | **81.3** | 77.6 |

Table 10: LMAO vs. PEFT baselines on OPT-1.3B (1000 examples). LOZO: Low-rank Zeroth-Order Stochastic Gradient Descent; SubZero: Random Subspace Zeroth-Order Optimization; DoRA: Dynamic Low-Rank Adaptation; AdaLoRA: Adaptive Budget Allocation for Parameter-Efficient Fine-Tuning; LMAO: Low-rank and Memory-efficient Zeroth-Order Alternating Optimization.

| Method | SST-2 | CB | BoolQ | COPA |
|---|---|---|---|---|
| LOZO | 86.0 | 72.0 | 62.4 | 69.0 |
| SubZero | 63.6 | 43.0 | **65.8** | 76.0 |
| AdaLoRA | 87.8 | 71.4 | 55.7 | 76.0 |
| DoRA | 87.3 | 69.4 | 60.6 | 75.0 |
| LMAO (ours) | **92.4** | **89.0** | 62.0 | **76.7** |

the four benchmark tasks. Except for isolated cases, it outperforms competing methods; even when a baseline holds a single-task advantage, LMAO's aggregate superiority remains evident, indicating stronger overall effectiveness and robustness.

## C CONVERGENCE ANALYSIS

### C.1 SYMBOL SPECIFICATION

The notation used in the derivation of the proofs is summarized in Table 11.

### C.2 PROOF OF LEMMA 4.3

*Proof.* Using the total differential formula, we have

$$\nabla\mathcal{L}(W_t; A_{t+1}, B_{t+1}) = [\nabla_{BA}\mathcal{L}(W_t; A_{t+1}, B_{t+1}), \nabla_W\mathcal{L}(W_t; A_{t+1}, B_{t+1})].$$

Table 11: Table of notations.

| Variable | Definition |
|---|---|
| $\mathcal{L}$ | The loss function. |
| $\mathcal{L}^*$ | The theoretical optimum of the objective function. |
| $A$ | Low-rank matrix in $\mathbb{R}^{r \times n}$. |
| $B$ | Low-rank matrix in $\mathbb{R}^{m \times r}$. |
| $W$ | The weight parameter matrix of a pre-trained model. |
| $\eta_{BA}$ | The learning rate of the low-rank module in Algorithm 1. |
| $\eta_W$ | The learning rate of the zeroth-order module in Algorithm 1. |
| $L_{BA}, L_W, L_{qua}$ | The Lipschitz constant. |
| $\mathcal{M}$ | A minibatch sample in SPSA-based sampling. |
| $|\mathcal{M}|$ | The number of samples $\mathcal{M}$ in a minibatch. |
| $\hat{\nabla}\mathcal{L}$ | The update gradient from algorithmic sampling. |
| $m, n$ | The dimensionality of the pre-trained model's parameter matrix. |
| $N$ | The number of samples drawn in SPSA. |
| $\alpha, \beta$ | The smoothness constant in the expected smoothness assumption. |
| $\mathbf{Sym}(\cdot)$ | The symmetrization function for tensors. |
| $\otimes$ | Tensor product. |

Under Assumption 4.1, we have (descent lemma)

$$
\begin{aligned}
&\mathcal{L}(W_{t+1}; A_{t+1}, B_{t+1}) \\
&\leqslant \mathcal{L}(W_t; A_t, B_t) + \nabla_{BA}^T \mathcal{L}(W_t; A_t, B_t)([B_{t+1}, A_{t+1}] - [B_t, A_t]) \\
&\quad + \frac{L_{BA}}{2} \big\|[B_{t+1}, A_{t+1}] - [B_t, A_t]\big\|_F^2 + \nabla_W^T \mathcal{L}(W_t; A_{t+1}, B_{t+1})(W_{t+1} - W_t) \\
&\quad + \frac{L_W}{2} \|W_{t+1} - W_t\|_F^2 \\
&\overset{(a)}{\leqslant} \mathcal{L}(W_t; A_t, B_t) - \eta_{BA}\big(\big\|\nabla_{BA}\mathcal{L}(W_t; A_t, B_t)\big\|_F^2 + \big\|\nabla_W\mathcal{L}(W_t; A_{t+1}, B_{t+1})\big\|_F^2\big) \\
&\quad + \frac{\eta_{BA}^2 L_{BA}}{2}\big\|\nabla_{BA}\mathcal{L}(W_t; A_t, B_t)\big\|_F^2 + \frac{\eta_{BA}^2 L_W}{2}\big\|\nabla_W\mathcal{L}(W_t; A_{t+1}, B_{t+1})\big\|_F^2 \\
&\overset{(b)}{\leqslant} \mathcal{L}(W_t; A_t, B_t) - \frac{\eta_{BA}}{2}\big(\big\|\nabla_{BA}\mathcal{L}(W_t; A_t, B_t)\big\|_F^2 + \big\|\nabla_W\mathcal{L}(W_t; A_{t+1}, B_{t+1})\big\|_F^2\big),
\end{aligned}
$$

where (a) is true because of the update rule of gradient descent in each block and Assumption 4.1, in (b) we used $\eta_{BA} \leqslant 1/L_{\max}$ and $L_{\max} := \max(L_{BA}, L_W)$. $\qquad \square$

### C.3 PROOF OF LEMMA 4.6

*Proof.* Using the equation equation 3 we note that

$$
\begin{aligned}
\mathbb{E}_{z \sim \mathcal{N}(0,I)}\big[\hat{\nabla}_W \mathcal{L}(W; A, B; \mathcal{M})\big] &= \mathbb{E}_{z \sim \mathcal{N}(0,I)}\big[zz^T \nabla_W \mathcal{L}(W; A, B; \mathcal{M})\big] \\
&= \nabla_W \mathcal{L}(W; A, B; \mathcal{M})\mathbb{E}_{z \sim \mathcal{N}(0,I)}\big[zz^T\big] \\
&= \nabla_W \mathcal{L}(W; A, B; \mathcal{M})\mathbb{E}_{z \sim \mathcal{N}(0,I)}I \\
&= \nabla_W \mathcal{L}(W; A, B; \mathcal{M}).
\end{aligned}
$$

$\square$

## C.4 PROOF OF LEMMA 4.7

*Proof.* We simplify the notation $\hat{\nabla}_W \mathcal{L}(W_t; A_t, B_t; \mathcal{M}|(x_i, y_i))$ as $\hat{\nabla}_W \mathcal{L}(W_t|((x_i, y_i))$, then we rewrite the expression as follows:

$$\mathbb{E}_{z \sim \mathcal{N}(0,I)}\big[\|\hat{\nabla}_W \mathcal{L}(W_t; A_t, B_t; \mathcal{M})\|_F^2\big]$$
$$= \mathbb{E}_{z \sim \mathcal{N}(0,I)}\big[\hat{\nabla}_W^T \mathcal{L}(W_t; A_t, B_t; \mathcal{M}) \cdot \hat{\nabla}_W \mathcal{L}(W_t; A_t, B_t; \mathcal{M})\big]$$
$$= \frac{1}{|\mathcal{M}|^2 N^2} \sum_{(x_1, y_1),(x_2, y_2) \in \mathcal{M}} \sum_{1 \leqslant i,j \leqslant N} \mathbb{E}\big[\hat{\nabla}_W \mathcal{L}(W_t|(x_1, y_1)) \hat{\nabla}_W^T \mathcal{L}(W_t|(x_2, y_2))\big]$$

Specifically, if $u$ and $v$ are two independent vectors, then we have

$$\mathbb{E}_{z_i, z_j}\big[z_i z_i^t u v^T z_j^T z_j\big] = uv^T,$$

when $i \neq j$ and

$$\mathbb{E}_{z_i}\big[z_i z_i^t u v^T z_i^T z_i\big] = \mathbb{E}_z\big[z^{\otimes 4}\big]\langle u, v\rangle$$
$$= \frac{3mn}{mn+2}\mathbf{Sym}(I^{\otimes 2})\langle u, v\rangle$$
$$= \frac{mn}{mn+2}u^T v I + \frac{mn}{mn+2}uv^T.$$

Therefore

$$\mathbb{E}_{z \sim \mathcal{N}(0,I)}\big[\hat{\nabla}_W^T \mathcal{L}(W_t; A_t, B_t; \mathcal{M}) \cdot \hat{\nabla}_W \mathcal{L}(W_t; A_t, B_t; \mathcal{M})\big]$$
$$= \frac{1}{|\mathcal{M}|^2} \sum_{(x_1, y_1)(x_2, y_2) \in \mathcal{M}} \big(\frac{N-1}{N} + \frac{2mn}{N(mn+2)}\big)$$
$$\times \mathbb{E}\big[(z_i z_i^T \hat{\nabla}_W \mathcal{L}(W_t|(x_1, y_1)))(z_j z_j^T \hat{\nabla}_W^T \mathcal{L}(W_t|(x_2, y_2)))^T\big]$$
$$+ \frac{mn}{N(mn+2)} \mathbb{E}\big[(\hat{\nabla}_W \mathcal{L}(W_t|(x_1, y_1)))^T (\hat{\nabla}_W^T \mathcal{L}(W_t|(x_2, y_2)))\big] I.$$

Note that when $(x_1, y_1) \neq (x_2, y_2)$, we have

$$\mathbb{E}\big[(\hat{\nabla}_W \mathcal{L}(W_t|(x_1, y_1)))^T (\hat{\nabla}_W^T \mathcal{L}(W_t|(x_2, y_2)))\big]$$
$$= \hat{\nabla}_W \mathcal{L}(W_t; A_t, B_t; \mathcal{M}) \hat{\nabla}_W^T \mathcal{L}(W_t; A_t, B_t; \mathcal{M}),$$

and when $(x_1, y_1) = (x_2, y_2)$, we have

$$\mathbb{E}\big[(\hat{\nabla}_W \mathcal{L}(W_t|(x_1, y_1)))^T (\hat{\nabla}_W^T \mathcal{L}(W_t|(x_2, y_2)))\big]$$
$$= \hat{\nabla}_W \mathcal{L}(W_t; A_t, B_t; \mathcal{M}) \hat{\nabla}_W^T \mathcal{L}(W_t; A_t, B_t; \mathcal{M}) + \sum_{\mathcal{M}},$$

then we get

$$\frac{1}{|\mathcal{M}|^2} \sum_{(x_1, y_1)(x_2, y_2) \in \mathcal{M}} \mathbb{E}\big[(\hat{\nabla}_W \mathcal{L}(W_t|(x_1, y_1)))^T (\hat{\nabla}_W^T \mathcal{L}(W_t|(x_2, y_2)))\big]$$
$$= \nabla_W \mathcal{L}(W_t; A_t, B_t) \nabla_W^T \mathcal{L}(W_t; A_t, B_t) + \sum_{\mathcal{M}}.$$

Summing these terms yields

$$\mathbb{E}\big[(\hat{\nabla}_W \mathcal{L}(W_t; A_t, B_t; \mathcal{M}))^T (\hat{\nabla}_W^T \mathcal{L}(W_t; A_t, B_t; \mathcal{M}))\big]$$
$$= \left(1 + \frac{mn}{N(mn+2)}\right) \left(\hat{\nabla}_W \mathcal{L}(W_t; A_t, B_t) \hat{\nabla}_W^T \mathcal{L}(W_t; A_t, B_t) + \frac{1}{|\mathcal{M}|}\sum_{\mathcal{M}}\right)$$
$$+ \frac{mn}{N(mn+2)} I \left(\|\hat{\nabla}_W \mathcal{L}(W_t; A_t, B_t)\|_F^2 + \frac{1}{|\mathcal{M}|}\mathbf{tr}(\sum_{\mathcal{M}}).\right).$$

From above, we obtain

$$\mathbb{E}\big[\|\hat{\nabla}_W \mathcal{L}(W_t; A_t, B_t; \mathcal{M})\|_F^2\big] = \frac{mn + N - 1}{N} \mathbb{E}\big[\|\hat{\nabla}_W \mathcal{L}(W_t; A_t, B_t)\|_F^2\big].$$

$\square$

## C.5 PROOF OF LEMMA 4.8

*Proof.* By the triangle inequality, we obtain

$$\|\nabla_W \mathcal{L}(W_t; A_t, B_t)\| \leqslant \|\nabla_W \mathcal{L}(W_t; A_{t+1}, B_{t+1}) - \nabla_W \mathcal{L}(W_t; A_t, B_t)\|$$
$$+ \|\nabla_W \mathcal{L}(W_t; A_{t+1}, B_{t+1})\|.$$

By the Mean inequality, we have

$$\|\nabla_W \mathcal{L}(W_t; A_t, B_t)\|_F^2 \leqslant 2\|\nabla_W \mathcal{L}(W_t; A_{t+1}, B_{t+1}) - \nabla_W \mathcal{L}(W_t; A_t, B_t)\|_F^2$$
$$+ 2\|\nabla_W \mathcal{L}(W_t; A_{t+1}, B_{t+1})\|_F^2.$$

Combining the updata procedure 2 with Assumption 4.1, we have

$$\|\nabla_W \mathcal{L}(W_t; A_t, B_t)\|_F^2 \leqslant 2L_{qua}^2 \|[A_{t+1}, B_{t+1}] - [A_t, B_t]\|_F^2 + 2\|\nabla_W \mathcal{L}(W_t; A_{t+1}, B_{t+1})\|_F^2$$
$$= 2L_{qua}^2 \eta_{BA}^2 \|\hat{\nabla}_{BA} \mathcal{L}(W_t; A_t, B_t)\|_F^2 + 2\|\nabla_W \mathcal{L}(W_t; A_{t+1}, B_{t+1})\|_F^2.$$

For a fixed $0 < \eta_{BA} < \dfrac{1}{L_{\max}}$, where $L_{\max} := \max(L_{BA}, L_W, L_{qua})$, the above expression reduces to:

$$\|\nabla_W \mathcal{L}(W_t; A_t, B_t)\|_F^2 \leqslant 2\left(\frac{L_{qua}^2}{L_{\max}^2}\|\hat{\nabla}_{BA} \mathcal{L}(W_t; A_t, B_t)\|_F^2 + \|\nabla_W \mathcal{L}(W_t; A_{t+1}, B_{t+1})\|_F^2\right).$$

Fixed $L_{\max}^2 \geqslant 2L_{qua}^2$, then the gradient norm of Algorithm 1 satisfies

$$\|\nabla \mathcal{L}(W_t; A_t, B_t)\|_F^2 = \|\nabla_W \mathcal{L}(W_t; A_t, B_t)\|_F^2 + \|\nabla_{BA} \mathcal{L}(W_t; A_t, B_t)\|_F^2$$
$$\leqslant 2\left(\frac{L_{qua}^2}{L_{\max}^2}\|\nabla_{BA} \mathcal{L}(W_t; A_t, B_t)\|_F^2 + \|\nabla_W \mathcal{L}(W_t; A_{t+1}, B_{t+1})\|_F^2\right)$$
$$+ \|\nabla_{BA} \mathcal{L}(W_t; A_t, B_t)\|_F^2$$
$$\leqslant \left(1 + 2\frac{L_{qua}^2}{L_{\max}^2}\right)\|\nabla_{BA} \mathcal{L}(W_t; A_t, B_t)\|_F^2 + 2\|\nabla_W \mathcal{L}(W_t; A_{t+1}, B_{t+1})\|_F^2$$
$$\leqslant \left(1 + 2\frac{L_{qua}^2}{L_{\max}^2}\right)\left(\|\nabla_{BA} \mathcal{L}(W_t; A_t, B_t)\|_F^2 + \|\nabla_W \mathcal{L}(W_t; A_{t+1}, B_{t+1})\|_F^2\right)$$
$$\leqslant 2\left(1 + \frac{L_{qua}}{L_{\max}}\right)^2\left(\|\nabla_{BA} \mathcal{L}(W_t; A_t, B_t)\|_F^2 + \|\nabla_W \mathcal{L}(W_t; A_{t+1}, B_{t+1})\|_F^2\right).$$

□

## C.6 PROOF OF THEOREM 4.9

*Proof.* We start from L-smoothness and use equation equation 2 and equation 4:

$$\mathcal{L}(W_{t+1}; A_{t+1}, B_{t+1}) \leqslant \mathcal{L}(W_t; A_t, B_t) + \langle \nabla_W \mathcal{L}(W_t; A_{t+1}, B_{t+1}), W_{t+1} - W_t \rangle$$
$$+ \langle \nabla_{BA} \mathcal{L}(W_t; A_t, B_t), ([B_{t+1}, A_{t+1}] - [B_t, A_t]) \rangle$$
$$+ \frac{L_W}{2}(\|[B_{t+1}, A_{t+1}] - [B_t, A_t]\|_F^2) + \frac{L_W}{2}\|W_{t+1} - W_t\|_F^2$$
$$\leqslant \mathcal{L}(W_t; A_t, B_t) - \eta_W \langle \nabla_W \mathcal{L}(W_t; A_{t+1}, B_{t+1}), \hat{\nabla}_W \mathcal{L}(W_t; A_{t+1}, B_{t+1}) \rangle$$
$$- \eta_{BA} \langle \nabla_{BA} \mathcal{L}(W_t; A_t, B_t), \nabla_{BA} \mathcal{L}(W_t; A_t, B_t) \rangle$$
$$+ \frac{L_{BA}}{2}\eta_{BA}^2 \|\nabla_{BA} \mathcal{L}(W_t; A_t, B_t)\|_F^2$$
$$+ \frac{L_W}{2}\eta_W^2 \|\hat{\nabla}_W \mathcal{L}(W_t; A_{t+1}, B_{t+1})\|_F^2.$$

Taking conditional expectation and combining Lemma 4.6, Lemma 4.7, we have

$$\mathbb{E}\big[\mathcal{L}(W_{t+1}; A_{t+1}, B_{t+1})|(W_t; A_t, B_t)\big]$$

$$\leqslant \mathcal{L}(W_t; A_t, B_t) - \eta_W \|\nabla_W \mathcal{L}(W_t; A_{t+1}, B_{t+1})\|_F^2 - \eta_{BA}\|\nabla_{BA}\mathcal{L}(W_t; A_t, B_t)\|_F^2$$

$$+ \frac{L_{BA}}{2}\eta_{BA}^2 \mathbb{E}[\|\nabla_{BA}\mathcal{L}(W_t; A_t, B_t)\|_F^2] + \frac{L_W}{2}\eta_W^2 \mathbb{E}[\|\hat{\nabla}_W \mathcal{L}(W_t; A_{t+1}, B_{t+1})\|_F^2]$$

$$\leqslant \mathcal{L}(W_t; A_t, B_t) - \eta_{\min}\left(\|\nabla_W \mathcal{L}(W_t; A_{t+1}, B_{t+1})\|_F^2 + \|\nabla_{BA}\mathcal{L}(W_t; A_t, B_t)\|_F^2\right)$$

$$+ \frac{L_{BA}}{2}\eta_{\max}^2 \mathbb{E}[\|\nabla_{BA}\mathcal{L}(W_t; A_t, B_t)\|_F^2]$$

$$+ \frac{L_W}{2}\eta_{\max}^2 \frac{mn + N - 1}{N}\mathbb{E}[\|\nabla_W \mathcal{L}(W_t; A_{t+1}, B_{t+1})\|_F^2].$$

Subtracting $\mathcal{L}^*$ from both sides of the inequality and using Assumption 4.2, we have

$$\mathbb{E}\big[\mathcal{L}(W_{t+1}; A_{t+1}, B_{t+1})|(W_t; A_t, B_t)\big] - \mathcal{L}^*$$

$$\leqslant \mathcal{L}(W_t; A_t, B_t) - \mathcal{L}^* - \eta_{\min}\left(\|\nabla_W \mathcal{L}(W_t; A_{t+1}, B_{t+1})\|_F^2 + \|\nabla_{BA}\mathcal{L}(W_t; A_t, B_t)\|_F^2\right)$$

$$+ \frac{L_{BA}}{2}\eta_{\max}^2 \left[2\alpha_1(\mathcal{L}(W_t; A_t, B_t) - \mathcal{L}^*) + \beta_1\|\nabla_{BA}\mathcal{L}(W_t; A_t, B_t)\|_F^2\right]$$

$$+ \frac{L_W}{2}\eta_{\max}^2 \frac{mn + N - 1}{N}\left[2\alpha_2(\mathcal{L}(W_t; A_{t+1}, B_{t+1}) - \mathcal{L}^*) + \beta_2\|\nabla_W \mathcal{L}(W_t; A_{t+1}, B_{t+1})\|_F^2\right].$$

Using the Assumpiton 4.1, we have

$$\mathcal{L}(W_t; A_{t+1}, B_{t+1}) - \mathcal{L}^* = \mathcal{L}(W_t; A_{t+1}, B_{t+1}) - \mathcal{L}(W_t; A_t, B_t) + \mathcal{L}(W_t; A_t, B_t) - \mathcal{L}^*$$

$$\leqslant \langle \nabla_{BA}\mathcal{L}(W_t; A_t, B_t), ([B_{t+1}, A_{t+1}] - [B_t, A_t])\rangle$$

$$+ \frac{L_{BA}}{2}\|[B_{t+1}, A_{t+1}] - [B_t, A_t]\|_F^2 + \mathcal{L}(W_t; A_t, B_t) - \mathcal{L}^*.$$

Define $\delta_{t+1} = \mathbb{E}\big[\mathcal{L}(W_{t+1}; A_{t+1}, B_{t+1})|[B_t, A_t]\big] - \mathcal{L}^*$ and combining the above equations, then we have

$$\delta_{t+1} \leqslant \delta_t - \eta_{\min}\left(\|\nabla_W \mathcal{L}(W_t; A_{t+1}, B_{t+1})\|_F^2 + \|\nabla_{BA}\mathcal{L}(W_t; A_t, B_t)\|_F^2\right)$$

$$+ \alpha_1 L_{BA}\eta_{\max}^2 \delta_t + \frac{1}{2}L_{BA}\eta_{\max}^2 \beta_1\|\nabla_{BA}\mathcal{L}(W_t; A_t, B_t)\|_F^2$$

$$+ \alpha_2 L_W \eta_{\max}^2 \frac{mn + N - 1}{N}\left(\mathcal{L}(W_t; A_{t+1}, B_{t+1}) - \mathcal{L}(W_t; A_t, B_t)\right)$$

$$+ \alpha_2 L_W \eta_{\max}^2 \frac{mn + N - 1}{N}\delta_t + \frac{1}{2}\beta_2 L_W \eta_{\max}^2\|\nabla_W \mathcal{L}(W_t; A_{t+1}, B_{t+1})\|_F^2$$

$$\leqslant \left(1 + \alpha_1 L_{BA}\eta_{\max}^2 + \alpha_2 L_W \eta_{\max}^2 \frac{mn + N - 1}{N}\right)\delta_t$$

$$- \eta_{\max}\left(\frac{\eta_{\min}}{\eta_{\max}} - \frac{1}{2}\beta_2 L_W \eta_{\max}^2 \frac{mn + N - 1}{N}\right)\|\nabla_W \mathcal{L}(W_t; A_{t+1}, B_{t+1})\|_F^2$$

$$- \eta_{\max}\left(\frac{\eta_{\min}}{\eta_{\max}} - \frac{1}{2}L_{BA}\eta_{\max}^2 \beta_1 - \alpha_2 L_W \eta_{\max}^2 \frac{mn + N - 1}{N}\left(-\eta_{\min} + \frac{L_{BA}}{2}\eta_{\max}^2\right)\right)$$

$$\times \|\nabla_{BA}\mathcal{L}(W_t; A_t, B_t)\|_F^2.$$

Combining the above equations, we have

$$\frac{1}{2000}\eta_{\max}\left(\|\nabla_W \mathcal{L}(W_t; A_{t+1}, B_{t+1})\|_F^2 + \|\nabla_{BA}\mathcal{L}(W_t; A_t, B_t)\|_F^2\right)$$

$$\leqslant \left(1 + \alpha_1 L_{BA}\eta_{\max}^2 + \alpha_2 L_W \eta_{\max}^2 \frac{mn + N - 1}{N}\right)\delta_t - \delta_{t+1},$$

where $\eta$ satisfies

$$\begin{cases} \dfrac{\eta_{\min}}{\eta_{\max}} - \dfrac{1}{2}\beta_2 L_W \eta_{\max}^2 \dfrac{mn + N - 1}{N} \geqslant \dfrac{1}{2000} \\ \dfrac{\eta_{\min}}{\eta_{\max}} - \dfrac{1}{2}L_{BA}\eta_{\max}^2 \beta_1 - \alpha_2 L_W \eta_{\max}\left(-\eta_{\min} + \dfrac{L_{BA}}{2}\eta_{\max}^2\right) \geqslant \dfrac{1}{2000} \\ -\eta_{\min} + \dfrac{L_{BA}}{2}\eta_{\max}^2 \leqslant \eta_{\max} \end{cases}$$

i.e.

$$\eta_{\max} \leqslant \min\left(\frac{N}{1000\beta_2 L_W(mn+N-1)}, \frac{1000N}{N\beta_1 L_{BA} + 2\alpha_2 L_W(mn+N-1)}, \frac{4}{L_{BA}}\right).$$

Let fix $p_{-1} > 0$ and define $p_t = p_{t-1}\left(1 + \alpha_1 L_{BA}\eta_{\max}^2 + \alpha_2 L_W\eta_{\max}^2\frac{mn+N-1}{N}\right)^{-1}$ for all $t \geqslant 0$. Multiplying by $\frac{p_t}{\eta_{\max}}$, we have

$$\frac{p_t}{2}\left(\|\nabla_W\mathcal{L}(W_t; A_{t+1}, B_{t+1})\|_F^2 + \|\nabla_{BA}\mathcal{L}(W_t; A_t, B_t)\|_F^2\right) \leqslant (p_{t-1}\delta_t - p_t\delta_{t+1}).$$

Summing up both sides as $t = 0, 1, 2, \cdots, T-1$, we have

$$\sum_{t=0}^{T-1}\frac{p_t}{2}\left(\|\nabla_W\mathcal{L}(W_t; A_{t+1}, B_{t+1})\|_F^2 + \|\nabla_{BA}\mathcal{L}(W_t; A_t, B_t)\|_F^2\right)$$
$$\leqslant (p_{t-1}\delta_t - p_t\delta_{t+1}) \leqslant p_{-1}\delta_0.$$

Define $P_T = \sum_{t=0}^{T-1} p_t$ and we have

$$P_T = \sum_{t=0}^{T-1} p_t \geqslant \sum_{t=0}^{T-1} \min\{p_0.p_1, \cdots, p_{T-1}\} = Tp_{T-1}$$

$$= \frac{Tp_{-1}}{\left(1 + \alpha_1 L_{BA}\eta_{\max}^2 + \alpha_2 L_W\eta_{\max}^2\frac{mn+N-1}{N}\right)^T}.$$

Dividing both sides by $P_T$ we have

$$\min_{0\leqslant t\leqslant T-1}\mathbb{E}\left[\|\nabla_W\mathcal{L}(W_t; A_{t+1}, B_{t+1})\|_F^2 + \|\nabla_{BA}\mathcal{L}(W_t; A_t, B_t)\|_F^2\right]$$

$$\leqslant \frac{1}{P_T}\sum_{t=0}^{T-1} p_t\left(\|\nabla_W\mathcal{L}(W_t; A_{t+1}, B_{t+1})\|_F^2 + \|\nabla_{BA}\mathcal{L}(W_t; A_t, B_t)\|_F^2\right)$$

$$\leqslant \frac{p_{-1}\delta_0}{\eta_{\max}P_T}.$$

Using the fact that $1 + x \leqslant \exp(x)$, we have

$$\left(1 + \alpha_1 L_{BA}\eta_{\max}^2 + \alpha_2 L_W\eta_{\max}^2\frac{mn+N-1}{N}\right)^T$$

$$\leqslant \exp T\left(\alpha_1 L_{BA}\eta_{\max}^2 + \alpha_2 L_W\eta_{\max}^2\frac{mn+N-1}{N}\right)$$

$$\leqslant \exp(1) \leqslant 3,$$

where the second inequality holds because $\eta_{\max} \leqslant \sqrt{N/\left(T(N\alpha_1 L_{BA} + \alpha_2 L_W(mn+N-1))\right)}$ by assumption. Combining the above equations, we have

$$\min_{0\leqslant t\leqslant T-1}\mathbb{E}\left[\|\nabla_W\mathcal{L}(W_t; A_{t+1}, B_{t+1})\|_F^2 + \|\nabla_{BA}\mathcal{L}(W_t; A_t, B_t)\|_F^2\right] \leqslant \frac{3(\mathcal{L}(W_0; A_0, B_0) - \mathcal{L}^*)}{\eta_{\max}T}.$$

By the Lemma 4.8, we have

$$\|\nabla\mathcal{L}(W_t; A_{t+1}, B_{t+1})\|_F^2$$

$$\leqslant 2\left(1 + \frac{L_{BA}}{L_{\max}}\right)^2\left(\|\nabla_{BA}\mathcal{L}(W_t; A_t, B_t)\|_F^2 + \|\nabla_W\mathcal{L}(W_t; A_{t+1}, B_{t+1})\|_F^2\right).$$

Taking expectations and minimizing over $t$ on both sides of the inequality, then we have

$$
\min_{0 \leqslant t \leqslant T-1} \mathbb{E}\left[\left\|\nabla \mathcal{L}(W_t; A_t, B_t)\right\|_F^2\right]
$$

$$
\leqslant \min_{0 \leqslant t \leqslant T-1} 2\left(1 + \frac{L_{BA}}{L_{\max}}\right)^2 \left(\mathbb{E}\left[\left\|\nabla_{BA}\mathcal{L}(W_t; A_t, B_t)\right\|_F^2\right] + \mathbb{E}\left[\left\|\nabla_W \mathcal{L}(W_t; A_{t+1}, B_{t+1})\right\|_F^2\right]\right)
$$

$$
\leqslant \frac{6(1 + \frac{L_{BA}}{L_{\max}})^2(\mathcal{L}(W_0; A_0, B_0) - \mathcal{L}^*)}{\eta_{\max} T}.
$$

$\square$

## D USE OF LLMS

LLMs are used for grammar checking and language polishing.

