# OpenReview forum: "Three Forward, One Backward: Memory-Efficient Full-Rank Fine-Tuning of Large Models via Extra Forward Passes"
_ICLR.cc/2026/Conference — ICLR 2026 Poster_

### Official Review · Reviewer_9oXR · 2025-10-18

**Soundness:** 2
**Presentation:** 3
**Contribution:** 2
**Rating:** 4
**Confidence:** 4

**Summary:**

Large language models have shown remarkable performance but fine-tuning them is computationally and memory-intensive due to growing model sizes. The motivation is to develop scalable fine-tuning methods that reduce resource demands while maintaining high performance on downstream tasks. Challenges include LoRA's suboptimal performance from low-rank constraints and MeZO's high variance and slow convergence from zeroth-order gradient estimates. The solution, LMAO, alternates low-rank matrix updates using backpropagation with zeroth-order updates on base parameters via forward passes, enabling full-rank fine-tuning with three forwards and one backward per iteration under memory constraints.

**Strengths:**

1. LMAO integrates the low-rank efficiency of LoRA with the memory-saving forward-only updates of MeZO. This hybrid approach achieves full-rank parameter updates without excessive memory use. It effectively reduces feature loss common in low-rank methods while controlling variance better than pure zeroth-order techniques.

2. Experimental results on RoBERTa-large demonstrate that LMAO outperforms baselines in few-shot and many-shot settings. It achieves accuracies close to or surpassing full fine-tuning on tasks like SST-2 and SNLI. This highlights its ability to optimize medium-sized models effectively across sentiment and inference tasks.

3. On larger OPT models up to 6.7B parameters, LMAO consistently ranks highest on most SuperGLUE tasks. It improves over LoRA on classification and multiple-choice problems like BoolQ and COPA. Such performance validates its scalability and generalization to autoregressive architectures.

4. The theoretical framework provides convergence guarantees under Lipschitz and expected smoothness assumptions. Descent lemmas and rate analyses ensure reliable optimization. This mathematical rigor strengthens the method's foundation beyond empirical evidence.

**Weaknesses:**

1. LMAO requires more forward passes than standard LoRA, increasing computational overhead. This can extend training time significantly on large datasets. Efficiency gains in memory do not fully offset the added flops.

2. Zeroth-order components introduce inherent noise despite alternation. This variance may hinder convergence on noisy or complex tasks. Smoother gradients from full backpropagation are absent in parts of the update.

3. Evaluations are limited to models up to 6.7B parameters. And the utilized evaluation datasets are outdated. Scaling issues could emerge with extreme sizes.

4. Comparisons omit advanced PEFT variants like sparse adapters. Only basic LoRA and MeZO are baselines. Broader benchmarking would clarify relative advantages.

5. Inference implications are undiscussed after training. Merging adapters into base weights could add latency. Deployment considerations are overlooked.

**Questions:**

See weaknesses.

---

> ### Author Response · Authors · 2025-11-26
>
> We sincerely appreciate the valuable time you dedicated to thoroughly reviewing our paper and providing constructive comments and professional guidance. Your feedback is insightful and highly relevant, offering important support for improving the quality of our work and strengthening the rigor of our presentation. We have carefully studied your suggestions and provide detailed explanations below:

---

> ### Author Response · Authors · 2025-11-26
>
> ### W1-Part1: LMAO requires more forward passes than standard LoRA, increasing computational overhead. This can extend training time significantly on large datasets. Efficiency gains in memory do not fully offset the added flops.
>
> We thank the reviewer for the insightful observation regarding the increased computational cost due to additional forward passes. We address this concern from three perspectives: the nature of the computational cost, the practical trade-off and methodological motivation, and supporting experiments. To this end, we have added supplemental experiments to analyze running time.
>
> * **Regarding the nature of the “three forward passes, one backward pass” computation**
>
>   * Although LMAO performs three forward passes per step, two of them are forward-only ZO steps, which do not require storing activations or computing gradients. Their cost is significantly lower than that of the forward pass used for backpropagation. Since backpropagation remains the dominant contributor to training cost, and LMAO performs only one backward pass per iteration (same as LoRA), the overall overhead per step is far below a naive “3×” multiplier.
>   * We acknowledge that, compared with standard LoRA, LMAO introduces additional forward passes per iteration. However, while LMAO does incur extra computation, it also yields substantial performance improvement. As shown in the updated experiments (Table 1 and Table 2), although LMAO requires more training time, this is a necessary cost for the performance gains. Overall, LMAO achieves the best or highly competitive performance across all four benchmark datasets, significantly outperforming the two baseline parameter-efficient fine-tuning methods.
>
> * **Practical trade-off and methodological motivation**
>
>   * LMAO is designed to achieve memory-efficient full-rank updates under strict GPU memory constraints. In such settings, FLOPs are typically not the primary bottleneck. Instead, LoRA and MeZO suffer from limited expressiveness or unstable training, respectively. LMAO achieves near–full-rank performance with memory usage similar to LoRA. For large models, this increased expressiveness translates into consistent and substantial accuracy improvements over LoRA and MeZO. Under the same memory constraints, this modest extra computation is a reasonable cost for stronger update capability.
>
>   * In practice, for large models, memory constraints are often more critical than FLOPs [1, 2]. Standard LoRA still requires storing activations for the entire network during backpropagation, which limits batch size and model scale under a fixed GPU budget. The MeZO component integrated in LMAO is lightweight in the forward process—it only requires applying random perturbations to model parameters and computing the loss, without additional memory overhead. This allows our method to improve accuracy while maintaining LoRA-level memory usage and only moderately increasing training time.
>
>     [1] Zhang, Longteng, et al. "Lora-fa: Memory-efficient low-rank adaptation for large language models fine-tuning." arXiv preprint arXiv:2308.03303 (2023).
>
>     [2] Malladi, Sadhika, et al. "Fine-tuning language models with just forward passes." Advances in Neural Information Processing Systems 36 (2023): 53038-53075.

---

> ### Author Response · Authors · 2025-11-26
>
> ### W1-Part2: LMAO requires more forward passes than standard LoRA, increasing computational overhead. This can extend training time significantly on large datasets. Efficiency gains in memory do not fully offset the added flops.
> * **Supplemental Experiments**
>
>   * **Table 1:** **Performance changes of MeZO under increased training steps.**
>
>     | Steps     | MeZO’s Training time（s）[CB-LMAO-1000step:2121.83588]       | MeZO’s Acc(CB-MeZO)   [CB-LMAO-1000step:0.89]                |
>     | --------- | ------------------------------------------------------------ | ------------------------------------------------------------ |
>     | 2000      | 832.48072                                                    | 0.79                                                         |
>     | 4000      | 1684.77869                                                   | 0.8                                                          |
>     | 6000      | 2524.55521                                                   | 0.76                                                         |
>     | 8000      | 3363.66275                                                   | 0.73                                                         |
>     | 10000     | 4201.08608                                                   | 0.72                                                         |
>     | 12000     | 5038.02708                                                   | 0.72                                                         |
>     | 14000     | 5871.50556                                                   | 0.72                                                         |
>     | 16000     | 6710.54874                                                   | 0.71                                                         |
>     | 18000     | 7548.92205                                                   | 0.72                                                         |
>     | 20000     | 8387.87202                                                   | 0.72                                                         |
>     | **Steps** | **MeZO’s** **Training time（s）[BoolQ-LMAO-1000step:462.41236]** | **MeZO’s** **Acc(BoolQ-MeZO)    [BoolQ-LMAO-1000step:0.65]** |
>     | 2000      | 992.27511                                                    | 0.618                                                        |
>     | 4000      | 1997.7152                                                    | 0.622                                                        |
>     | 6000      | 3006.93512                                                   | 0.626                                                        |
>     | 8000      | 4012.14557                                                   | 0.612                                                        |
>     | 10000     | 5019.61636                                                   | 0.622                                                        |
>     | 12000     | 6029.28202                                                   | 0.612                                                        |
>     | 14000     | 7037.22374                                                   | 0.612                                                        |
>     | 16000     | 8044.59016                                                   | 0.612                                                        |
>     | 18000     | 9053.8758                                                    | 0.61                                                         |
>     | 20000     | 10057.76712                                                  | 0.608                                                        |

---

> ### Author Response · Authors · 2025-11-26
>
> ### W1-Part3: LMAO requires more forward passes than standard LoRA, increasing computational overhead. This can extend training time significantly on large datasets. Efficiency gains in memory do not fully offset the added flops.
> * **Table 2: Comparison of extended LoRA training time vs. LMAO on the CB dataset**
>
>     | Dataset | Steps (LMAO) | Time (LMAO) | Acc (LMAO) | Steps (LoRA) | Time (LoRA) | Acc (LoRA) |
>     | ------- | ------------ | ----------- | ---------- | ------------ | ----------- | ---------- |
>     | CB      | 1000         | 2123.16616  | 0.93       | 1000         | 578.50517   | 0.82       |
>     |         | 1000         | 2123.16616  | 0.93       | 2000         | 1166.34468  | 0.85       |
>     |         | 1000         | 2123.16616  | 0.93       | 3000         | 1752.862653 | 0.85       |
>     |         | 1000         | 2123.16616  | 0.93       | 4000         | 2336.09745  | 0.85       |

---

> ### Author Response · Authors · 2025-11-26
>
> ### W2: Zeroth-order components introduce inherent noise despite alternation. This variance may hinder convergence on noisy or complex tasks. Smoother gradients from full backpropagation are absent in parts of the update.
>
> We appreciate the reviewer’s concern regarding the higher variance introduced by zeroth-order components compared to gradient-based updates.
>
> * We agree that gradient-based methods generally provide smoother parameter updates. However, as discussed in W1, relying solely on full backpropagation significantly increases memory consumption, thereby imposing stronger hardware constraints during fine-tuning. Our LMAO method preserves the benefits of zeroth-order optimization while mitigating the high-variance gradient estimates inherent in MeZO. Both theoretical and empirical evidence supporting this claim are provided in the paper.
>
> * Specifically, we provide a detailed proof of Theorem 4.9, which demonstrates that the iterations of LMAO satisfy an \( O(1/T) \) convergence bound in terms of the average gradient norm. In addition, our experiments show no observable negative impact on convergence caused by the zeroth-order components. On RoBERTa-large, LMAO consistently outperforms all gradient-free baselines (MeZO and its variants) across six classification tasks under both \( k = 16 \) and \( k = 512 \) settings, achieving performance comparable to or even exceeding that of full-parameter fine-tuning.
>
> * To provide further evidence, we include additional experiments comparing LMAO and MeZO under equal wall-clock time. From the perspective of convergence speed, LMAO typically reaches convergence in fewer iterations than MeZO. As a result, the overall convergence time is not increased but instead shortened, indicating that the noise introduced by zeroth-order components does not negatively affect the optimization process. (See Table 1 and Table 2 in W1.)

---

> ### Author Response · Authors · 2025-11-26
>
> ### W3: Evaluations are limited to models up to 6.7B parameters. And the utilized evaluation datasets are outdated. Scaling issues could emerge with extreme sizes.
>
> We appreciate the reviewer for emphasizing the importance of comprehensive comparisons. We have supplemented our evaluations with additional results on OPT-13B for the CB dataset.
>
> * **Supplementary results on OPT-13B**.
>
>   * Due to limitations in time and computational resources, we were only able to obtain results on the CB dataset at this stage, and we report them below.
>
>     **Table 3: Comparison of methods on OPT-13B**
>
>     | Methods | Acc (CB)    |
>     | ------- | ----------- |
>     | LoRA    | 0.660714286 |
>     | MeZO    | 0.482142857 |
>     | LMAO    | 0.678571429 |
>
> The results indicate that our method retains its advantage even at larger model scales. Given the limited rebuttal period, we will continue updating the evaluation results on newer datasets over the next few days.

---

> ### Author Response · Authors · 2025-11-26
>
> ### W4: Comparisons omit advanced PEFT variants like sparse adapters. Only basic LoRA and MeZO are baselines. Broader benchmarking would clarify relative advantages.
>
> We appreciate the reviewer’s suggestion to include a broader set of baselines for a more comprehensive comparison.
>
> * It is important to clarify that our work introduces a new alternating optimization framework for PEFT, along with theoretical guarantees for its feasibility and convergence. Under this theoretical framework, the LoRA component and the MeZO component can be optimized independently to achieve stronger performance improvements. The theory demonstrates extensibility, and the experiments with the core components validate the framework’s effectiveness. Therefore, our initial comparisons focus on these two fundamental components.
>
> * To further address the reviewer’s comment, we additionally conduct experiments with several advanced and widely used PEFT variants. The results show that our method clearly outperforms LOZO, SubZero, AdaLoRA and DoRA.
>
>   **Table 4: Comparison of mainstream PEFT methods on four benchmark datasets**
>
>   | Method \ Dataset | SST-2 | Copa | CB   | BoolQ |
>   |------------------|-------|------|------|-------|
>   | LMAO             | 92.4  | 76.7 | 89.0 | 62.0  |
>   | LOZO             | 86.0  | 69.0 | 72.0 | 62.4  |
>   | SubZero          | 63.6  | 76.0 | 43.0 | 65.8  |
>   | AdaLoRA          | 87.8  | 76.0 | 71.4 | 55.7  |
>   | DoRA             | 87.3  | 75.0 | 69.4 | 60.6  |
>
>   Overall, LMAO achieves the best or near-best performance across all four benchmark datasets, substantially outperforming existing mainstream PEFT methods. These results further validate the effectiveness and generalization ability of our method’s design.

---

> ### Author Response · Authors · 2025-11-26
>
> ### W5: Inference implications are undiscussed after training. Merging adapters into base weights could add latency. Deployment considerations are overlooked.
>
> We appreciate the reviewer’s valuable comments regarding inference-time implications and deployment considerations. We provide clarifications below concerning adapter merging and potential inference latency:
>
> * First, similar to standard LoRA-based methods, LMAO also supports efficiently merging the adapter back into the base model weights after training. This merging is performed only once before deployment and incurs negligible computational cost. Therefore, it introduces no additional latency during inference, and the runtime complexity of LMAO at inference time is identical to that of the original model.
>
> * Secondly, we will provide further explanations regarding deployment considerations in the revised version. Since LMAO adopts the same parameterization structure as LoRA, the deployment workflow is fully compatible with existing LoRA-based inference and serving frameworks. No modifications to the inference pipeline are required, and thus the engineering cost of deploying LMAO is equivalent to that of LoRA, without any added overhead.
>
> * We will include a more detailed discussion on inference overhead and deployment workflow in the Appendix, and we hope these additions will address the reviewer’s concerns.

---

### Official Review · Reviewer_N9p4 · 2025-10-24

**Soundness:** 3
**Presentation:** 3
**Contribution:** 3
**Rating:** 8
**Confidence:** 3

**Summary:**

This paper introduces a novel PEFT method for large language models. The proposed approach integrates the advantages of LoRA and MeZO by alternating between low-rank gradient updates and zeroth-order updates. Specifically, LMAO performs three forward passes and one backward pass per iteration, enabling full-rank parameter updates under stringent memory constraints. Overall, the paper presents a reasonably innovative idea; however, it would benefit from a broader set of experiments to better demonstrate its generalization capability across different model architectures. Moreover, I would like to see its performance on MMMU.

**Strengths:**

[1]. The authors provide open-source code and describe the experimental setup thoroughly, which enhances the transparency and reproducibility of the work.

[2]. The proposed method, LMAO, demonstrates strong empirical performance, outperforming or matching full fine-tuning on multiple NLP benchmarks, showing both effectiveness and efficiency.

[3]. The paper is clearly written, well-organized, and easy to follow. The motivation, methodology, and results are presented in a logical and coherent manner.

**Weaknesses:**

[1]. The method reduces memory usage during the zeroth-order phase, but the backward pass for low-rank updates still incurs considerable memory cost. This may limit scalability when applied to very large models or long-sequence tasks.

[2]. Although the paper claims good scalability, experiments are only conducted on models up to OPT-2.7B. Evaluating larger models such as LLaMA 3 (13B or 70B) or Qwen2.5 would better support the scalability claim.

[3]. The experiments focus only on text classification and inference tasks. It remains unclear whether the method generalizes well to generative or multimodal settings such as MMMU.

**Questions:**

[1]. The performance on multimodality settings?

[2]. The performance on DyLoRA or AdaLoRA?

---

> ### Author Response · Authors · 2025-11-26
>
> We sincerely appreciate the valuable time you dedicated to thoroughly reviewing our paper and providing constructive and professional feedback. Your comments are insightful and highly relevant, helping us improve the quality and rigor of our work. We have carefully considered your suggestions and provide detailed responses below.

---

> ### Author Response · Authors · 2025-11-26
>
> ### W1: The method reduces memory usage during the zeroth-order phase, but the backward pass for low-rank updates still incurs considerable memory cost. This may limit scalability when applied to very large models or long-sequence tasks.
>
> We thank the reviewer for the insightful comments regarding memory efficiency in LMAO. We address this concern from three perspectives: clarification of the paper’s core contribution, justification of the memory trade-off, and empirical results.
>
> - **Clarifying the core contribution of the paper**:
>
>   - We acknowledge that LMAO introduces some memory overhead on top of MeZO. However, we emphasize that the goal of LMAO is not to further reduce memory usage beyond LoRA or MeZO. Rather, it is designed to combine two memory-efficient fine-tuning algorithms and improve performance without significantly increasing memory consumption.
>
>   - While MeZO enjoys lower memory usage compared to LoRA, it generally underperforms in terms of accuracy and suffers from convergence instability due to the high variance in zeroth-order gradient estimation. LoRA, on the other hand, offers stable gradient information through backpropagation.
>
>   - Our LMAO framework effectively combines the strengths of both approaches: LoRA stabilizes optimization with gradient-based updates, while MeZO compensates for the representational limitations of LoRA’s low-rank updates by providing full-rank parameter updates. This synergy is both rational and effective.
>
>   - Furthermore, we propose a new alternating optimization framework for PEFT with theoretical guarantees on feasibility and convergence. Theoretically, LMAO alternates between LoRA and MeZO updates. Therefore, its memory consumption is bounded by the maximum peak memory of either component, rather than the sum. Since zeroth-order updates in MeZO do not require storing activations or gradients, they do not incur additional memory overhead.
>
>   - The framework is also extensible and not restricted to LoRA and MeZO. Both components can be independently optimized under this design, allowing for further performance improvements with future variants.
>
> - **Justifying the additional memory trade-off**:
>
>   - The extra memory cost incurred by LMAO mainly comes from minor communication and caching switches during alternation between LoRA and MeZO, which accounts for less than 5% of total training time. This minimal cost is well justified by the performance gain.
>
>   - While LMAO introduces moderate computational overhead compared to standalone LoRA or MeZO, the efficiency and optimization strength of the method are evident in our results: under the same number of training steps (e.g., 1000), LMAO consistently outperforms both LoRA and MeZO across multiple datasets.
>
> - **Experimental validation**:
>
>   - To address your concern on memory consumption, we measured peak GPU memory usage and performance across six datasets (CB, BoolQ, SST-2, Copa, ReCoRd, and MultiRC). Results show that LMAO does not significantly increase memory usage compared to the two components. In some cases, it even consumes less memory than LoRA while achieving better performance.
>
>
>   - **Table 1: Memory usage and performance comparison of LMAO vs. LoRA and MeZO**
>
>     | Dataset | Method | Peak Memory (GB) | Accuracy |
>     | ------- | ------ | ---------------- | -------- |
>     | Copa    | LoRA   | 8.31             | 74.0     |
>     |         | MeZO   | 7.70             | 74.7     |
>     |         | LMAO   | 8.31             | 76.7     |
>     | MultiRC | LoRA   | 67.01            | 49.0     |
>     |         | MeZO   | 11.93            | 56.3     |
>     |         | LMAO   | 64.49            | 56.7     |
>     | ReCoRd  | LoRA   | 16.64            | 71.4     |
>     |         | MeZO   | 8.78             | 72.0     |
>     |         | LMAO   | 15.83            | 72.1     |
>     | BoolQ   | LoRA   | 38.48            | 55.6     |
>     |         | MeZO   | 8.75             | 61.4     |
>     |         | LMAO   | 23.08            | 62.0     |
>     | CB      | LoRA   | 65.05            | 46.4     |
>     |         | MeZO   | 10.99            | 67.9     |
>     |         | LMAO   | 65.05            | 69.0     |
>     | SST-2   | LoRA   | 12.62            | 58.1     |
>     |         | MeZO   | 7.90             | 92.2     |
>     |         | LMAO   | 12.62            | 92.4     |
>
>     These results confirm that LMAO does not introduce additional memory overhead beyond the individual components, yet achieves notable improvements in performance.

---

> ### Author Response · Authors · 2025-11-26
>
> ### W2: Although the paper claims good scalability, experiments are only conducted on models up to OPT-2.7B. Evaluating larger models such as LLaMA 3 (13B or 70B) or Qwen2.5 would better support the scalability claim.
>
> * In our paper, we applied LMAO to OPT-1.3B, OPT-2.7B, and OPT-6.7B, and observed that LMAO consistently maintained its performance advantages as model size increased.
>
> * These results suggest that our method retains stable and scalable performance when the number of model parameters grows. In addition, we have begun to test LMAO on even larger-scale models. Due to time constraints during the rebuttal phase, we plan to report more results over the coming days, including evaluations on OPT-13B.
>
> * At this stage, we have obtained preliminary results on one dataset (CB) using OPT-13B, as shown below. More results will be updated soon.
>
>   **Table 2: Performance comparison on OPT-13B**
>
>   | Method | Acc (CB)    |
>   | ------ | ----------- |
>   | LoRA   | 0.660714286 |
>   | MeZO   | 0.482142857 |
>   | LMAO   | 0.678571429 |
>
>   These initial results further demonstrate that LMAO maintains strong scalability even on larger models, and we are continuing to validate its performance on additional datasets and architectures (e.g., LLaMA 3 and Qwen2.5) in our ongoing experiments.

---

> ### Author Response · Authors · 2025-11-26
>
> ### Q1 (W3): The performance on multimodality settings?
>
> We sincerely thank the reviewer for the insightful suggestion regarding performance in multimodal settings. Extending our method to multimodal architectures is indeed an important and promising future direction.
>
> * Due to limitations in time and computational resources, we have not yet conducted comprehensive validation on Multimodal Large Language Models. However, we believe the core principles of our method, including the advantages of full-rank updates and improved optimization convergence, are systematically transferable and provide a solid foundation for future exploration in multimodal settings.
>
> * We plan to systematically explore this extension in future work and truly appreciate the reviewer for highlighting this valuable direction.
>
> ---

---

> ### Author Response · Authors · 2025-11-26
>
> ### Q2: The performance on DyLoRA or AdaLoRA?
>
> We appreciate the reviewer’s interest in how our method compares with more recent PEFT variants such as DoRA and AdaLoRA. In response, we have conducted additional experiments, including not only LoRA-based variants but also comparisons with another zeroth-order method, to provide a more comprehensive and convincing benchmark.
>
> * **Table 3: Comparison of mainstream PEFT methods across four benchmark datasets**
>
>   | Method \ Dataset | SST-2 | Copa | CB   | BoolQ |
>   |------------------|-------|------|------|--------|
>   | LMAO             | 92.4  | 76.7 | 89.0 | 62.0   |
>   | LOZO             | 86.0  | 69.0 | 72.0 | 62.4   |
>   | SubZero          | 63.6  | 76.0 | 43.0 | 65.8   |
>   | AdaLoRA          | 87.8  | 76.0 | 71.4 | 55.7   |
>   | DoRA             | 87.3  | 75.0 | 69.4 | 60.6   |
>
> Overall, LMAO achieves the best or near-best performance across all four benchmark datasets, significantly outperforming existing state-of-the-art PEFT methods. These results strongly support the effectiveness and generalization ability of our method’s design.

---

> ### Author Response · Authors · 2025-12-04
>
> ## W2(Part-2): Although the paper claims good scalability, experiments are only conducted on models up to OPT-2.7B. Evaluating larger models such as LLaMA 3 (13B or 70B) or Qwen2.5 would better support the scalability claim.
>
> We sincerely thank the reviewers and the AC for their patience and thoughtful feedback throughout the review process. Due to earlier limitations in computational resources and time, we were unable to provide comprehensive evaluations on larger-scale models.
>
> - To better address the reviewers’ concerns, we have recently conducted additional experiments under a unified setup using the OPT-13B model, and we now include results for LoRA, MeZO, and LMAO on the BoolQ and Copa benchmarks. The results are summarized below:
>
> | Methods | Acc(CB)     | Acc(BoolQ) | Acc(Copa) |
> | ------- | ----------- | ---------- | --------- |
> | LoRA    | 0.660714286 | 0.616      | 0.85      |
> | MeZO    | 0.482142857 | 0.634      | 0.80      |
> | LMAO    | 0.678571429 | 0.659      | 0.87      |
>
> - As shown in the extended experiments, when scaling up to the 13B model, LMAO continues to demonstrate clear and consistent advantages across multiple datasets: it achieves the best performance on CB and Copa, and also outperforms other baselines on BoolQ. These results further validate the core claim presented in the main paper—LMAO not only delivers substantial improvements on small- and medium-scale models, but also maintains strong scalability and robustness at larger model sizes.
>
> These additional experiments reinforce the effectiveness of our approach and provide further support for the central contributions of the paper from the perspective of larger parameter scales.

---

### Official Review · Reviewer_6rvB · 2025-10-27

**Soundness:** 3
**Presentation:** 3
**Contribution:** 2
**Rating:** 6
**Confidence:** 4

**Summary:**

This paper introduces LMAO (Low-rank and Memory-efficient Zeroth-Order Alternating Optimization), a novel fine-tuning method for large language models that combines the strengths of LoRA and MeZO in an alternating optimization framework. LMAO performs gradient-based updates on low-rank LoRA matrices using one forward and one backward pass, followed by memory-efficient zeroth-order updates on full-rank base model weights using two additional forward passes. This "three forward, one backward" approach ensures base model parameters receive full-rank updates, overcoming LoRA's primary limitation while maintaining memory efficiency. The paper provides theoretical convergence analysis for this alternating scheme and demonstrates through extensive experiments on RoBERTa-large and OPT models up to 6.7B parameters that LMAO consistently outperforms LoRA and MeZO, achieving performance competitive with and sometimes superior to full fine-tuning.

**Strengths:**

* The main strength is the creative combination of LoRA and MeZO. It uses the precise gradients from backpropagation for the low-rank part while leveraging the memory efficiency of ZO for the full-rank part, achieving the best of both worlds.
* The method's effectiveness is demonstrated on models of increasing scale, from RoBERTa-large (350M) up to OPT-6.7B. This suggests the approach is a scalable solution for future, even larger models.
* The authors provide a formal convergence analysis for their algorithm. This adds a layer of theoretical rigor that is often missing in purely empirical papers and builds confidence in the method's stability.

**Weaknesses:**

* The most significant weakness is the computational cost. The "three forward, one backward" approach inherently requires more computation per iteration than LoRA (one forward, one backward) or MeZO (two forward). The paper acknowledges this as "training inefficiency" in the conclusion but fails to provide a quantitative analysis of the wall-clock training time. This is a crucial piece of information for practitioners to assess the practical trade-offs.
*  While the method is motivated by memory efficiency, the paper does not present direct measurements of peak GPU memory usage during training. A table or plot comparing the memory consumption of LMAO against FT, LoRA, and MeZO would make the "memory-efficient" claim much more concrete and impactful.
*  By combining two different optimizers, LMAO introduces more hyperparameters that require tuning (e.g., learning rate for LoRA $\eta_{BA}$, learning rate for ZO $\eta_W$, perturbation scale $\epsilon$, LoRA rank $r$ and alpha). While a sensitivity analysis for $r$ and $\alpha$ is included, a more in-depth discussion on the tuning strategy and the sensitivity to the relative learning rates would be beneficial for the method's practical adoption.

**Questions:**

1.  Could the authors provide a quantitative comparison of the wall-clock training time for LMAO against the baselines (especially LoRA and FT) for a representative task? For example, reporting the time to complete 1K training steps for OPT-1.3B on one of the SuperGLUE tasks would be very informative for understanding the practical computational trade-offs.
2.  To further strengthen the claims, could the authors provide a table or figure detailing the peak GPU memory consumption (e.g., in GB) of LMAO compared to FT, LoRA, and MeZO under an identical experimental setup (e.g., same model, batch size, and task)?
3.  LMAO uses separate learning rates for the ZO update ($\eta_W$) and the LoRA update ($\eta_{BA}$). Could the authors comment on the sensitivity of the model's performance to the ratio of these two learning rates? Are there any heuristics or best practices you discovered for setting them?
4. In Algorithm 1 Line 9, the step computes a projection and then updates each parameter by that scalar times its corresponding perturbation entry z. That is a rank-1 direction in the vectorized parameter space (one random direction), not a full-rank update. The paper repeatedly asserts “ensuring full-rank updates,” which is incorrect per-iteration; at best, many different random directions across iterations can span the full space in expectation. Please fix the claim and its implications (e.g., “reduces feature loss”) or use multiple independent directions per step.

**Details Of Ethics Concerns:**

No ethics concerns.

---

> ### Author Response · Authors · 2025-11-26
>
> We sincerely appreciate the valuable time you dedicated to thoroughly reviewing our paper and providing constructive comments and professional guidance. Your feedback is insightful and highly relevant, offering important support for improving the quality of our work and strengthening the rigor of our presentation. We have carefully studied your suggestions and provide detailed explanations below:

---

> ### Author Response · Authors · 2025-11-26
>
> ### Q1 (W1)-Part1: Could the authors provide a quantitative comparison of the wall-clock training time for LMAO against the baselines (especially LoRA and FT) for a representative task?
>
> We thank the reviewer for raising this important question. We fully agree that computational efficiency is critical for practical applications. Below, we clarify the computational characteristics of LMAO and provide corresponding quantitative results.
>
> * **Regarding the nature of the “three forward passes, one backward pass” computation**
>
>   * Although LMAO performs three forward passes per training step, two of them are forward-only ZO steps that do not require storing activations and do not involve gradient computation. Their cost is therefore significantly lower than the forward pass used for backpropagation. Since backpropagation remains the main contributor to training cost, LMAO performs only one backward pass per iteration—the same as LoRA.
>
> * **Practical trade-off and methodological motivation**
>
>   * The goal of LMAO is to enable memory-efficient full-rank updates under strict GPU memory constraints. In such settings, FLOPs are typically not the primary bottleneck. Instead, LoRA and MeZO suffer from limited expressiveness and unstable training, respectively. LMAO achieves near–full-rank performance while maintaining memory usage comparable to LoRA. For large models, this improved expressiveness leads to consistent and substantial accuracy gains over LoRA and MeZO. Under the same memory constraints, this moderate computational overhead is a reasonable price for stronger update capability.
>
> * **Quantitative training time results**
>
>   * We agree that reporting wall-clock cost is important. Under the same number of training iterations, we compare the training time of LMAO, LoRA, and FT on OPT-1.3B. Based on these results, we also conducted further experiments extending the training of LoRA and MeZO to observe convergence behaviors.
>
>     **Table 1:** **Performance changes of MeZO under increased training steps.**
>
>     | Training Steps       | Training Time (s) [CB-LMAO-1000step:2121.83588] | Acc (CB-MeZO) [CB-LMAO-1000step:0.89] |
>     |----------------------|--------------------------------------------------|----------------------------------------|
>     | 2000                 | 832.48072                                        | 0.79                                   |
>     | 4000                 | 1684.77869                                       | 0.8                                    |
>     | 6000                 | 2524.55521                                       | 0.76                                   |
>     | 8000                 | 3363.66275                                       | 0.73                                   |
>     | 10000                | 4201.08608                                       | 0.72                                   |
>     | 12000                | 5038.02708                                       | 0.72                                   |
>     | 14000                | 5871.50556                                       | 0.72                                   |
>     | 16000                | 6710.54874                                       | 0.71                                   |
>     | 18000                | 7548.92205                                       | 0.72                                   |
>     | 20000                | 8387.87202                                       | 0.72                                   |
>     | **Training Steps**   | **Training Time (s) [BoolQ-LMAO-1000step:462.41236]** | **Acc (BoolQ-MeZO) [BoolQ-LMAO-1000step:0.65]** |
>     | 2000                 | 992.27511                                        | 0.618                                  |
>     | 4000                 | 1997.7152                                        | 0.622                                  |
>     | 6000                 | 3006.93512                                       | 0.626                                  |
>     | 8000                 | 4012.14557                                       | 0.612                                  |
>     | 10000                | 5019.61636                                       | 0.622                                  |
>     | 12000                | 6029.28202                                       | 0.612                                  |
>     | 14000                | 7037.22374                                       | 0.612                                  |
>     | 16000                | 8044.59016                                       | 0.612                                  |
>     | 18000                | 9053.8758                                        | 0.61                                   |
>     | 20000                | 10057.76712                                      | 0.608                                  |

---

> ### Author Response · Authors · 2025-11-26
>
> ### Q1 (W1)-Part2: Could the authors provide a quantitative comparison of the wall-clock training
> * **Table 2: Extending LoRA training on CB (comparison with LMAO)**
>
> | Dataset | Steps (LMAO) | Time (LMAO) | acc (LMAO) | Steps (LoRA) | Time (LoRA)     | Acc (LoRA) |
> |---------|----------------|-------------|------------|----------------|------------------|------------|
> | CB      | 1000           | 2123.16616  | 0.93       | 1000           | 578.50517        | 0.82       |
> |         | 1000           | 2123.16616  | 0.93       | 2000           | 1166.34468       | 0.85       |
> |         | 1000           | 2123.16616  | 0.93       | 3000           | 1752.862653      | 0.85       |
> |         | 1000           | 2123.16616  | 0.93       | 4000           | 2336.09745       | 0.85       |
>
> * Even when MeZO is trained for 20× more iterations, our method still maintains a significant advantage.
> * Even with a substantially extended training budget, LoRA cannot surpass LMAO. Under equal or larger wall-clock budgets, LoRA’s convergence accuracy remains limited, while LMAO achieves higher performance in significantly shorter training time. This observation aligns with our theoretical analysis, which predicts LMAO’s stronger optimization efficiency and more robust convergence behavior.
> * Due to the limited time for the rebuttal, the experiments on FT are still ongoing. We will add the FT results to the table in the coming days.

---

> ### Author Response · Authors · 2025-11-26
>
> ### Q2 (W2): To further strengthen the claims, could the authors provide a table or figure detailing the peak GPU memory consumption (e.g., in GB) of  LMAO compared to FT, LoRA, and MeZO under an identical experimental setup (e.g., same model, batch size, and task)?
>
> We appreciate the reviewer’s insightful suggestion. We have supplemented the corresponding results in the table below. It is worth noting that the primary goal of our method is to improve performance without increasing the memory footprint of existing PEFT methods, rather than to further reduce memory usage. LoRA and MeZO each have their own memory–performance trade-offs, and LMAO integrates both to achieve a balanced combination of efficiency and expressiveness.
>
> - **Table 3: Memory consumption and performance comparison between LMAO and its components**
>
>   | Dataset | Method | Peak Memory (GB) | Acc |
>   |:--------|:-------|:------------------|:-----------:|
>   | Copa    | LoRA   | 8.31              |    74.0     |
>   |         | MeZO   | 7.70              |    74.7     |
>   |         | LMAO   | 8.31              |    76.7     |
>   | MultiRC | LoRA   | 67.01             |    49.0     |
>   |         | MeZO   | 11.93             |    56.3     |
>   |         | LMAO   | 64.49             |    56.7     |
>   | ReCoRd  | LoRA   | 16.64             |    71.4     |
>   |         | MeZO   | 8.78              |    72.0     |
>   |         | LMAO   | 15.83             |    72.1     |
>   | BoolQ   | LoRA   | 38.48             |    55.6     |
>   |         | MeZO   | 8.75              |    61.4     |
>   |         | LMAO   | 23.08             |    62.0     |
>   | CB      | LoRA   | 65.05             |    46.4     |
>   |         | MeZO   | 10.99             |    67.9     |
>   |         | LMAO   | 65.05             |    69.0     |
>   | SST-2   | LoRA   | 12.62             |    58.1     |
>   |         | MeZO   | 7.90              |    92.2     |
>   |         | LMAO   | 12.62             |    92.4     |
>
> * The results demonstrate that LMAO does not introduce additional memory overhead beyond its two components.
>   Yet, it achieves significant performance gains, validating the effectiveness of combining gradient-based and gradient-free updates under the same memory budget.

---

> ### Author Response · Authors · 2025-11-26
>
> ### Q3 (W3): Could the authors comment on the sensitivity of the model's performance to the ratio of these two learning rates? Are there any heuristics or best practices you discovered for setting them?
>
> We appreciate the reviewer’s insightful question regarding the sensitivity of LMAO to hyperparameter settings. In fact, our paper already includes sensitivity analyses on the key hyperparameters—namely the rank and the perturbation scale—which are core parameters of the underlying PEFT components. Their demonstrated stability further supports the reliability of the proposed alternating optimization framework.
>
> * Regarding the learning rates, our alternating update mechanism helps the optimization escape local minima and improves global search ability, thereby reducing sensitivity to fine-grained learning rate tuning. The additional experiments provided below confirm this behavior. LMAO maintains stable performance even under different learning-rate configurations.
>
> * **Additional experiments:**
>
>   **Table 4:Sensitivity analysis of hyperparameters related to the learning rate**
>
>   | Dataset | LR ($\eta_{BA}$, $\eta_W$) | Acc         | LR ($\eta_{BA}$, $\eta_W$) | Acc         | LR ($\eta_{BA}$, $\eta_W$) | Acc         |
>   | ------- | -------------------------------------- | ----------- | -------------------------------------- | ----------- | -------------------------------------- | ----------- |
>   | CB      | 1e-5, 1e-5                             | 0.571428571 | 1e-5, 1e-6                             | 0.678571429 | 1e-6, 1e-5                             | 0.517857143 |
>   | BoolQ   | 1e-5, 1e-5                             | 0.636       | 1e-5, 1e-6                             | 0.636       | 1e-6, 1e-5                             | 0.662       |
>   | SST-2   | 1e-5, 1e-5                             | 0.498853211 | 1e-5, 1e-6                             | 0.498853211 | 1e-6, 1e-5                             | 0.48853211  |
>   | Copa    | 1e-5, 1e-5                             | 0.83        | 1e-5, 1e-6                             | 0.87        | 1e-6, 1e-5                             | 0.81        |
>
> * These results demonstrate that LMAO is relatively robust to the choice and ratio of learning rates, and does not require delicate tuning to achieve strong performance.

---

> ### Author Response · Authors · 2025-11-26
>
> ### Q4: That is a rank-1 direction in the vectorized parameter space (one random direction), not a full-rank update.
>
> We sincerely appreciate the reviewer’s insightful comment regarding our discussion of full-rank updates, which indeed touches upon one of the core conceptual points of our paper. We clarify this from the perspectives of the definition of “full-rank update” used in our paper and  the theoretical interpretation behind it.
>
> * **Definition of full-rank updates in our paper**
>
>   * In LoRA, the parameter update is decomposed into two low-rank matrices (W = W₀ + BA). This dramatically reduces the number of trainable parameters—from billions in the base model to only millions or fewer. However, the resulting low-rank update space limits its expressiveness and prevents the learning of more complex features. This limitation motivates our alternating optimization framework LMAO, which integrates LoRA with MeZO.
>   * In contrast, the zeroth-order update operates as a full-parameter update, since it perturbs all model parameters in a random direction and estimates the directional derivative based on the change in loss. While each individual perturbation direction is a vector, the key point is that the optimization is conducted directly in the full parameter space—not restricted to a low-rank subspace as in LoRA. Over iterations, these random directions span a high-dimensional update space, effectively enabling full-rank optimization.
>   * MeZO alone, however, suffers from high-variance gradient estimates, which may hurt stability. LoRA complements this by providing a structured and stable low-variance gradient component. By alternating the two, LMAO ensures both expressiveness and stable convergence.

---

### Official Review · Reviewer_sMQJ · 2025-10-28

**Soundness:** 2
**Presentation:** 3
**Contribution:** 1
**Rating:** 2
**Confidence:** 4

**Summary:**

LMAO introduces a novel alternating optimization framework for fine-tuning large language models, combining LoRA's low-rank efficiency with MeZO's memory-friendly zeroth-order updates. By performing three forward passes and one backward pass per iteration, it enables full-rank adaptations while minimizing memory use and feature loss, achieving convergence guarantees and superior performance over baselines like LoRA and MeZO on tasks such as SuperGLUE with models like OPT and RoBERTa.

**Strengths:**

This paper features solid writing with a clear, logical structure and concise language, allowing quick understanding of innovations and results. The theory is rigorous, supported by convergence proofs and rate analyses under assumptions like Lipschitz smoothness for reliability. Experiments are thorough, covering models like RoBERTa and OPT series across tasks like SuperGLUE, including few-shot setups, ablations, sensitivity analyses, and comparisons to baselines such as LoRA and MeZO.

**Weaknesses:**

While the paper is innovative, it has some drawbacks. Though it targets resource-constrained fine-tuning, LMAO doesn't match MeZO's ultra-low, inference-level memory efficiency due to the required backward pass, which adds noticeable memory overhead. It emphasizes full-rank updates for improved expressiveness, but the performance gains over LoRA are often marginal(see Ablation Study), making the benefits feel underwhelming. Plus, the extra forward passes introduce significant computational costs, potentially making it inefficient for long sequences or large datasets, and rendering it somewhat niche or even a bit of a half-measure compared to simpler options like LoRA.

**Questions:**

1) the MeZO baseline results in Table 2 on certain datasets seem unusually low—for instance, on SST-2 and CB—which raises questions about their reliability and might suggest inconsistencies in implementation or evaluation setup[1][2][3].

2) The experiments fall short by missing key baselines; they should include comparisons with advanced zeroth-order optimization algorithms designed to mitigate MeZO's high variance issues, such as SubZero[2] and LoZO[3]. Additionally, contrasts with LoRA variants that emulate full-rank updates—like DoRA[4]—would be valuable. Overall, the evaluations need a more comprehensive analysis that balances fine-tuning performance against memory usage and computational time overhead.

[1] Zhang, Yihua, et al. "Revisiting Zeroth-Order Optimization for Memory-Efficient LLM Fine-Tuning: A Benchmark." International Conference on Machine Learning. 2024.

[2] Yu, Ziming, et al. "Zeroth-order fine-tuning of llms in random subspaces." Proceedings of the IEEE/CVF International Conference on Computer Vision. 2025.

[3] Chen, Yiming, et al. "Enhancing Zeroth-order Fine-tuning for Language Models with Low-rank Structures." The Thirteenth International Conference on Learning Representations.

[4] Liu, Shih-Yang, et al. "Dora: Weight-decomposed low-rank adaptation." Forty-first International Conference on Machine Learning. 2024.

**Details Of Ethics Concerns:**

None.

---

> ### Author Response · Authors · 2025-11-26
>
> We sincerely appreciate the time and effort you have dedicated to reviewing our paper and for providing constructive and insightful comments. Your feedback is both precise and highly valuable, offering important guidance for improving the clarity, rigor, and overall quality of our work. We have carefully considered your suggestions and provide detailed responses below.

---

> ### Author Response · Authors · 2025-11-26
>
> ### W1:  While the paper is innovative, it has some drawbacks. Though it targets resource-constrained fine-tuning, LMAO doesn't match MeZO's ultra-low, inference-level memory efficiency due to the required backward pass, which adds noticeable memory overhead.
>
> We thank the reviewer for the comments regarding the memory efficiency of LMAO. We address this concern from three perspectives—our core contributions, the justification of the incurred overhead, and additional experiments—to clarify the design and efficiency of our framework.
>
> * Clarification of the core contributions:
>   * We acknowledge that LMAO introduces some additional memory overhead compared to MeZO. However, it is important to emphasize that the goal of LMAO is not to further reduce memory consumption relative to these baselines. Instead, our objective is to combine two memory-efficient fine-tuning paradigms and achieve significant performance improvements without substantially increasing memory usage.
>   * Although MeZO is more memory-efficient than LoRA, its performance is generally inferior to LoRA. Moreover, due to the inherently high variance of zero-order gradient estimation, MeZO often suffers from unstable convergence in practice.
>   * In addition, we introduce a new PEFT alternating optimization framework and provide theoretical justification supporting its feasibility and convergence. Our analysis shows that LMAO performs alternating optimization between a low-rank component (LoRA) and a zero-order component (MeZO). Consequently, the memory usage is determined by the maximum of the two individual memory peaks. Since the zero-order component does not introduce additional memory overhead, the overall memory consumption of LMAO remains comparable to that of the underlying methods, without incurring extra memory cost.
>   * The theoretical analysis also indicates that the proposed framework is extensible and not limited to the basic forms of LoRA and MeZO. Under this framework, the LoRA component and the MeZO component can be further improved independently, providing a principled way to achieve additional performance gains.
>
> * Justification of the additional overhead:
>   * The additional memory overhead mainly comes from minor communication and buffer switching during the alternation between LoRA and MeZO. This cost accounts for less than 5% of the total training time, and we consider it a reasonable trade-off given the performance gains achieved. Although our method introduces some additional computation compared to vanilla LoRA and MeZO, the experiments clearly justify this overhead: under the same training budget (e.g., 1000 steps), our approach consistently achieves significantly better performance across multiple datasets, demonstrating higher training efficiency and stronger optimization capability.
>
> * Experimental validation:
>
>   * To address the concern regarding the significance of memory overhead, we conducted experiments on six datasets (CB, BoolQ, SST-2, Copa, ReCoRd, MultiRC). The results show that combining the two methods does not introduce noticeable additional memory consumption; in some cases, our approach even achieves higher performance with lower memory usage compared to LoRA. In practice, the combined method does not lead to any substantial increase in memory cost. We provide a summary of representative results below:
>
>   * **Table 1: Memory usage and performance comparison between LMAO and its two component methods.**
>
>     | TASK    | Method | Memory overhead (peak, GB) | Acc(%) |
>     | :------ | :----- | :------------------------- | :----: |
>     | Copa    | LoRA   | 8.31                       |  74.0  |
>     |         | MeZO   | 7.70                       |  74.7  |
>     |         | LMAO   | 8.31                       |  76.7  |
>     | MultiRC | LoRA   | 67.01                      |  49.0  |
>     |         | MeZO   | 11.93                      |  56.3  |
>     |         | LMAO   | 64.49                      |  56.7  |
>     | ReCoRd  | LoRA   | 16.64                      |  71.4  |
>     |         | MeZO   | 8.78                       |  72.0  |
>     |         | LMAO   | 15.83                      |  72.1  |
>     | BoolQ   | LoRA   | 38.48                      |  55.6  |
>     |         | MeZO   | 8.75                       |  61.4  |
>     |         | LMAO   | 23.08                      |  62.0  |
>     | CB      | LoRA   | 65.05                      |  46.4  |
>     |         | MeZO   | 10.99                      |  67.9  |
>     |         | LMAO   | 65.05                      |  69.0  |
>     | SST-2   | LoRA   | 12.62                      |  58.1  |
>     |         | MeZO   | 7.90                       |  92.2  |
>     |         | LMAO   | 12.62                      |  92.4  |
>
>     The results show that LMAO does not incur additional memory usage beyond that of its two components, while achieving substantial performance improvements.

---

> ### Author Response · Authors · 2025-11-26
>
> ### W2:  It emphasizes full-rank updates for improved expressiveness, but the performance gains over LoRA are often marginal(see Ablation Study), making the benefits feel underwhelming.
>
> We thank the reviewer for the comments regarding the full-rank aspect of our method. We clarify this point from two perspectives: the specific meaning of “full-rank” in our formulation and the supporting ablation experiments.
>
> * First, the term “full-rank update” in the paper refers to weight updates that are not constrained to any low-rank subspace in the matrix sense. Since MeZO operates directly on all model parameters and performs updates in the full parameter space, it effectively compensates for the representational limitations caused by LoRA’s low-rank updates.
> * Second, we conducted a systematic experimental evaluation of the methods, and the results are summarized in Table 1. The findings show that, under the same memory budget as LoRA and without introducing additional computational or storage overhead, LMAO achieves substantial performance improvements across multiple benchmark datasets. LMAO consistently outperforms both LoRA and MeZO by a large margin, further validating the effectiveness and advantages of our design.

---

> ### Author Response · Authors · 2025-11-26
>
> ### W3: Plus, the extra forward passes introduce significant computational costs, potentially making it inefficient for long sequences or large datasets, and rendering it somewhat niche or even a bit of a half-measure compared to simpler options like LoRA.
>
> We sincerely thank the reviewer for the valuable feedback. Regarding the concern about the additional forward passes potentially increasing computational cost, we provide further clarification below.
>
> * As the reviewer noted, our method introduces a small amount of additional computation on top of LoRA. However, our experiments show that this overhead accounts for only a minor portion of the total training cost, while yielding consistently significant performance gains across multiple datasets. The improvements are stable and substantially exceed those of LoRA and other baselines.
>
> * Therefore, we believe that this modest additional cost is reasonable and well-justified in practice, especially in scenarios where higher model quality is desired. We have also added more detailed experiments and ablation studies in the paper to clarify the trade-off between computational overhead and performance benefits.
>
> * In addition, to further address the reviewer’s concerns regarding training cost, we conducted an additional comparison. Specifically, we extended the training schedule of LoRA on the CB dataset to evaluate its performance under more extensive training. The results are shown in the table below:
>
>   **Table 2: Comparison on the CB dataset with extended LoRA training (vs. LMAO)**
>
>   | TASK | Steps(LMAO) | Training time(LMAO) | Acc（LMAO） | Steps(LoRA) | Training time（LoRA） | Acc(LoRA) |
>   | ---- | ----------- | ------------------- | ----------- | ----------- | --------------------- | --------- |
>   | CB   | 1000        | 2123.16616          | 0.93        | 1000        | 578.50517             | 0.82      |
>   |      | 1000        | 2123.16616          | 0.93        | 2000        | 1166.34468            | 0.85      |
>   |      | 1000        | 2123.16616          | 0.93        | 3000        | 1752.862653           | 0.85      |
>   |      | 1000        | 2123.16616          | 0.93        | 4000        | 2336.09745            | 0.85      |
>
>   These results show that even with substantially extended training, LoRA still fails to surpass LMAO. Under comparable or even larger training budgets, LoRA remains limited in its achievable accuracy, whereas LMAO reaches higher performance within significantly fewer training steps. This observation aligns with our theoretical analysis, which suggests that LMAO provides more reliable guarantees in terms of optimization efficiency and convergence behavior.

---

> ### Author Response · Authors · 2025-11-26
>
> ### Q1: The MeZO baseline results in Table 2 on certain datasets seem unusually low—for instance, on SST-2 and CB—which raises questions about their reliability and might suggest inconsistencies in implementation or evaluation setup。
>
> We thank the reviewer for the question regarding our experimental setup and fully understand the concern about fairness. Both the main paper and the appendix provide complete details of the training configurations, and all experiments were conducted strictly following these specifications.
>
> * Regarding the observation that MeZO performs lower on some datasets compared to its original paper, this is primarily because the MeZO paper allows the method to run substantially more iterations than other baselines.
>
> * To ensure fairness, we reduced the number of MeZO training iterations in our experiments, which naturally leads to lower performance. As stated in the MeZO paper (Section 3.2, page 6), MeZO is claimed to be more computationally efficient, and thus they set its training steps to be 20× larger than other methods (20,000 steps for MeZO vs. 1,000 for others). However, when running the released code, we found that its actual training throughput was lower than reported, so we reduced its training steps for an equitable comparison.
>
> * To fully address the reviewer’s concern, we additionally conducted experiments where MeZO is allowed to train for up to 20,000 steps. The results show that even with 20× more iterations, MeZO still fails to match the performance of our method, and LMAO remains significantly better.
>
>   **Table 3: Performance of MeZO with increased training steps**
>
>   | Steps     | MeZO’s Training time（s）[CB-LMAO-1000step:2121.83588]       | MeZO’s Acc(CB-MeZO)   [CB-LMAO-1000step:0.89]                |
>   | --------- | ------------------------------------------------------------ | ------------------------------------------------------------ |
>   | 2000      | 832.48072                                                    | 0.79                                                         |
>   | 4000      | 1684.77869                                                   | 0.8                                                          |
>   | 6000      | 2524.55521                                                   | 0.76                                                         |
>   | 8000      | 3363.66275                                                   | 0.73                                                         |
>   | 10000     | 4201.08608                                                   | 0.72                                                         |
>   | 12000     | 5038.02708                                                   | 0.72                                                         |
>   | 14000     | 5871.50556                                                   | 0.72                                                         |
>   | 16000     | 6710.54874                                                   | 0.71                                                         |
>   | 18000     | 7548.92205                                                   | 0.72                                                         |
>   | 20000     | 8387.87202                                                   | 0.72                                                         |
>   | **Steps** | **MeZO’s** **Training time（s）[BoolQ-LMAO-1000step:462.41236]** | **MeZO’s** **Acc(BoolQ-MeZO)    [BoolQ-LMAO-1000step:0.65]** |
>   | 2000      | 992.27511                                                    | 0.618                                                        |
>   | 4000      | 1997.7152                                                    | 0.622                                                        |
>   | 6000      | 3006.93512                                                   | 0.626                                                        |
>   | 8000      | 4012.14557                                                   | 0.612                                                        |
>   | 10000     | 5019.61636                                                   | 0.622                                                        |
>   | 12000     | 6029.28202                                                   | 0.612                                                        |
>   | 14000     | 7037.22374                                                   | 0.612                                                        |
>   | 16000     | 8044.59016                                                   | 0.612                                                        |
>   | 18000     | 9053.8758                                                    | 0.61                                                         |
>   | 20000     | 10057.76712                                                  | 0.608                                                        |

---

> ### Author Response · Authors · 2025-11-26
>
> ### Q2:The experiments fall short by missing key baselines; they should include comparisons with advanced zeroth-order optimization algorithms designed to mitigate MeZO's high variance issues, such as SubZero and LoZO.
>
> We thank the reviewer for encouraging us to conduct a more comprehensive evaluation of LMAO. In response, we have added experiments comparing against LOZO, SubZero, AdaLoRA, and DoRA.
>
> * The results show that our method consistently outperforms several recent zero-order or low-rank optimization approaches.
>
> * The updated results are presented below:
>
>   **Table 4: Performance comparison of mainstream parameter-efficient fine-tuning methods on four benchmark datasets**
>
>   | Method/Task | SST-2 | Copa | CB   | BoolQ |
>   | ----------- | ----- | ---- | ---- | ----- |
>   | LMAO        | 92.4  | 76.7 | 89.0 | 62.0  |
>   | LOZO        | 86.0  | 69.0 | 72.0 | 62.4  |
>   | SubZero     | 63.6  | 76.0 | 43.0 | 65.8  |
>   | AdaLoRA     | 87.8  | 76.0 | 71.4 | 55.7  |
>   | DoRA        | 87.3  | 75.0 | 69.4 | 60.6  |
>
>   Overall, LMAO achieves the best or near-best performance across all four benchmark datasets, clearly outperforming existing mainstream parameter-efficient fine-tuning methods. This provides strong evidence for the effectiveness and generalization capability of our approach.
>
>   In addition, our theoretical framework indicates that LMAO is highly extensible. We plan to incorporate more advanced low-rank and zero-order techniques within this framework in future work to further enhance performance.

---

### Author Response · Authors · 2025-11-29

# Summary of Revisions to the Manuscript

We sincerely thank all reviewers for their constructive feedback.

Specifically, we thank the reviewers for recognizing our contributions, including:

* The theoretical framework provides convergence guarantees under Lipschitz continuity and expected smoothness assumptions. The descent lemma and rate analysis ensure reliable optimization, and this mathematical rigor strengthens the method’s foundations beyond empirical evidence.
* A creative integration of LoRA and MeZO: the low-rank component leverages exact backpropagated gradients, while the full-rank component exploits the memory efficiency of zeroth-order (ZO) updates—achieving the best of both worlds.
* The method’s effectiveness is demonstrated across increasingly large models, from RoBERTa-large (350M) to OPT-6.7B, indicating a scalable solution for even larger models in the future.
* The paper is clearly written, well organized, and easy to follow; the motivation, methodology, and results are presented in a logical and coherent manner.

### Revisions to the Main Text (per first-round feedback)

1. **Ablation studies (Sec. 5.3).** We integrated the additional experiments introduced during the discussion stage into the main paper, enriched the narrative, and updated the figure accordingly. The original data were not modified; we only appended the new experimental results and reflected them in the tables/figures.
   * Page 9, Sec. 5.3: expanded the exposition to articulate the ablation conclusions.
   * Page 9, Sec. 5.3: refined and clarified the key findings from the ablations.
2. **Memory analysis.**
   * Page 10, Sec. 5.4: added a new subsection dedicated to GPU memory analysis, accompanied by the corresponding supplemental table.
   * Page 10, Sec. 5.5: incorporated the memory experiments conducted during the discussion stage into the main text and added a concise discussion of the findings.

### Revisions to the Appendix (per first-round feedback)

1. **Supplemental experiments.**
   * Page 15: moved the original supplemental content to Appendix B.1 for clarity.
   * Page 17: added new experimental results as Appendix B.2.
2. **Hyperparameter settings.**
   * Appendix B.1, Table 7: augmented with detailed hyperparameter configurations used in the supplemental experiments

---

### Author Response · Authors · 2025-12-02

# **Summary after the Discussion Period**

We greatly appreciate the reviewers' detailed feedback and constructive suggestions. Below, we provide a concise summary of how the revised manuscript addresses all the substantial concerns, including the memory consumption of the methods, the MeZO experimental results, the scalability of the models, and the comparative experiments. For the convenience of the AC, we have summarized and organized the reviewers' feedback and our responses in bullet points.

## Acknowledgment of the Manuscript's Strengths

* The theory is rigorously developed, supported by convergence proofs and rate analysis, with assumptions such as Lipschitz smoothness that ensure reliability. (sMQJ, 6rvB, 9oXR)
* The writing is solid, with a clear structure and logical flow, and the language is concise, making the innovations and results easy to understand. (sMQJ, N9p4)
* The experiments are extensive and rigorous. (sMQJ, N9p4)
* The method demonstrates clear advantages, along with scalability and reproducibility. (6rvB, 9oXR)

## Responses to Weaknesses and Issues

The four reviewers raised the following key concerns:

1. **Performance of MeZO on [OPT-1.3B] is significantly lower compared to other works.**
   * The models used in our work differ from those in other studies, leading to different performance outcomes.
   * The MeZO experiments mentioned in the manuscript had different settings in terms of the number of iterations (1000 steps for other experiments, but 20,000 steps for MeZO). We have aligned the number of steps in the experiments.
   * We ran the MeZO experiment for 20,000 steps, and we have provided performance, memory, and time statistics to substantiate our claims.

2. **Memory consumption is not significantly reduced.**
   * We clarified that one of the core contributions of our work is to improve performance without increasing memory consumption.
   * We provided detailed statistics on the memory consumption, runtime, and performance from previous experiments, demonstrating that while the energy consumption has not increased, performance has been significantly improved.

3. **The comparison experiments are not comprehensive enough.**
   * Our LMAO algorithm is an optimization framework consisting of alternating low-rank LoRA components and zero-order MeZO components. To ensure effective comparison, we have added additional experiments.
   * We have included variants of LoRA, such as DoRA and AdaLoRA, as well as zero-order methods like SubZero, and LoRA-zero hybrids such as LOZO. The results confirm that our method continues to exhibit significant advantages.

4. **Questions regarding model scalability.**
   * We presented experiments on OPT-1.3B, OPT-2.7B, and OPT-6.7B, demonstrating that LMAO maintains significant advantages as the model size increases.
   * Due to computational resource constraints, we have also included results for OPT-13B on select datasets. The results show that LMAO continues to exhibit strong performance.

5. **Clarification on full-rank updates.**
   * We provided a detailed explanation of the term "rank," specifically that the updates to the model weights are full-rank. In detail:
     The term "full-rank updates" refers to weight updates that are full-rank in the matrix sense, meaning the updates are not constrained by a low-rank assumption. Since the MeZO optimization directly affects all model weights, it updates the entire parameter space, partially compensating for the limited expressiveness of low-rank updates in LoRA.

6. **Concerns about the training time of LMAO.**
   * We re-ran our experiments, including those for LoRA and MeZO, and recorded the time consumption. Additionally, we conducted an extra 6000 steps of training. The results indicate that although our method requires additional training time, the performance improvement is substantial. Furthermore, after the extra 6000 steps, the algorithm shows better convergence, confirming the validity of the results.

---

### Meta-Review · Area_Chair_vqFe · 2026-01-07

**Summary:**

The proposed alternating framework effectively combines low-rank gradient-based updates with zeroth-order full-parameter updates, addressing known limitations of both LoRA and MeZO. The reviewers initially raised several concerns regarding the positioning and empirical validation of the proposed LMAO framework:

**Memory vs. Efficiency Trade-off:** A primary concern (raised by Reviewers sMQJ and others) was that LMAO does not achieve the ultra-low memory footprint of pure Zeroth-Order (ZO) methods like MeZO because it still requires storing activations for the gradient-based backward pass. Reviewers questioned whether the "memory-efficient" claim was overstated, given this overhead.

**Baselines and Comparisons:** Reviewers noted a lack of comprehensive comparisons with relevant state-of-the-art methods, specifically requesting baselines for advanced ZO methods (e.g., SubZero, LoZO) and full-rank LoRA variants (e.g., DoRA). Also, there was significant skepticism regarding the reported performance of the MeZO baseline, which some reviewers (Reviewer sMQJ) found unusually low compared to established benchmarks, raising questions about implementation correctness or hyperparameter fairness (e.g., iteration counts).

**Computational Overhead and Scalability:** Reviewers flagged the increased training time caused by the "three forward passes" requirement, questioning if the performance gains justified the FLOPs increase. Also, there were requests for validation on larger-scale models beyond the initially reported sizes (e.g., requesting OPT-13B).

Overall, mos

**Reviewer Concerns:**

The authors responded thoroughly to reviewers’ concerns by adding missing baselines, memory and runtime analyses, and larger-scale experiments, which together strengthen the empirical validity and clarify the practical trade-offs. There are no significant outstanding concerns.

**Reviewer Scores:**

Reviewer sMQJ, who gave the most critical score, likely would have raised the score, and other reviewers would likely have maintained or raised their scores.

---

### Decision · Program_Chairs · 2026-01-26

Accept (Poster)